# QFFT, Question-Free Fine-Tuning for Adaptive Reasoning

**Wanlong Liu**[*1,2], **Junxiao Xu**[*2], **Fei Yu**[2], **Yukang Lin**[2], **Ke Ji**[2], **Wenyu Chen**[1]
**Yan Xu**[3†], **Yasheng Wang**[3], **Lifeng Shang**[3], **Benyou Wang**[2†]
[1] University of Electronic Science and Technology of China, Chengdu, China
[2] The Chinese University of Hong Kong, Shenzhen
[3] Huawei Noah's Ark Lab

## Abstract

Recent advancements in Long Chain-of-Thought (CoT) reasoning models have improved performance on complex tasks, but they suffer from overthinking, which generates redundant reasoning steps, especially for simple questions. This paper revisits the reasoning patterns of Long and Short CoT models, observing that the Short CoT patterns offer concise reasoning efficiently, while the Long CoT patterns excel in challenging scenarios where the Short CoT patterns struggle. To enable models to leverage both patterns, we propose Question-Free Fine-Tuning (QFFT), a fine-tuning approach that removes the input question during training and learns exclusively from Long CoT responses. This approach enables the model to adaptively employ both reasoning patterns: it prioritizes the Short CoT patterns and activates the Long CoT patterns only when necessary. Experiments on various mathematical datasets demonstrate that QFFT reduces average response length by more than 50%, while achieving performance comparable to Supervised Fine-Tuning (SFT). Additionally, QFFT exhibits superior performance compared to SFT in noisy, out-of-domain, and low-resource scenarios. QFFT is publicly available at https://github.com/LWL-cpu/Question-Free-Fine-Tuning.

## 1 Introduction

Recent advancements in Long CoT reasoning models, such as OpenAI o1 [1] and DeepSeek-R1 [2], have significantly improved performance on complex tasks, such as mathematics and coding [3, 4, 5, 6, 7]. These improvements are largely attributed to the test-time scaling paradigm, where models generate Long CoT, consuming more tokens, to effectively simulate human-like deep thinking behavior like self-reflection, error correction, and exploration of multiple solution strategies. Building on this success, a growing body of research has focused on distillation methods that transfer the Long CoT reasoning abilities of OpenAI o1 and DeepSeek-R1 into smaller LLMs, achieving notable performance gains [8, 9, 10, 2].

However, recent studies [11, 12] have identified a critical limitation in current Long CoT models (e.g., DeepSeek-R1): they often exhibit **overthinking**, generating unnecessarily complex or redundant reasoning steps even for simple problems. As a result, models distilled from these Long CoT models tend to inherit this drawback, leading to inefficiencies during inference.

To mitigate this issue, recent *Long-to-Short* methods [13, 14, 15, 16, 17] explore compressing the length of Long CoT responses. However, these methods incur substantial additional training costs while achieving only limited reductions in token usage.

---

[*]Wanlong Liu and Junxiao Xu contributed to this work equally.
[†]Yan Xu and Benyou Wang are the corresponding authors.

39th Conference on Neural Information Processing Systems (NeurIPS 2025).

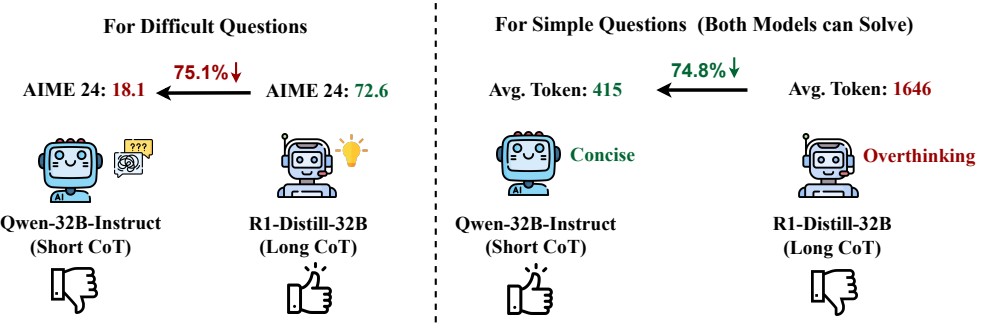

Figure 1: Comparison of short and Long CoT reasoning patterns. The left subplot shows that the Long CoT patterns perform better on difficult questions, while the right subplot demonstrates that, for simple questions solvable by both, the Short CoT patterns provide significantly more concise answers.

In this paper, we compare the reasoning patterns of Short CoT models and Long CoT models: (1) For efficiency: Short CoT models provide concise and efficient reasoning, whereas Long CoT models often generate lengthy outputs, which typically involve unnecessarily complex or redundant steps for simple problems. (2) For effectiveness: Short CoT models struggle with more difficult problems due to their simplistic reasoning patterns, while Long CoT models, with their reflective reasoning, demonstrate clear advantages in such scenarios. This contrast motivates a natural question: **Can we design models that adaptively select between Short CoT and Long CoT reasoning patterns?** Such models would ideally combine the strengths of both reasoning patterns—employing Short CoT reasoning for simple problems to enhance efficiency, and leveraging Long CoT reasoning for difficult problems to pursue effectiveness.

We revisit current Long CoT Supervised Fine-Tuning (SFT): models are trained on large-scale *(question, Long CoT response)* pairs, suffering from overthinking issues. Inspired by [18], we hypothesize that this issue arises because SFT enforces the mapping from questions to Long CoT responses, overriding the model's original Short CoT patterns and causing it to apply Long CoT reasoning indiscriminately—even when concise reasoning would suffice. To design models that could leverage both Short CoT and Long CoT patterns for a better balance between efficiency and effectiveness, we set two goals: (1) preserve the model's default Short CoT reasoning patterns, and (2) enable it to acquire Long CoT patterns that trigger reflective behaviors when facing uncertainties or errors. To achieve these objectives, we propose **Q**uestion-**F**ree **F**ine-**T**uning (QFFT), a fine-tuning method that discards input questions and only fine-tunes Long CoT responses. This approach avoids overriding the Short CoT patterns while still enabling the model to learn reflective reasoning patterns of Long CoT.

Our extensive experiments demonstrate that the QFFT method adaptively integrates both reasoning patterns: it employs Short CoT reasoning for simple problems to enhance efficiency, and leverages Long CoT reasoning for more challenging problems. Further analysis reveals that the QFFT model prioritizes Short CoT patterns by default, transitioning to reflective Long CoT reasoning when encountering errors or uncertainties. Notably, QFFT achieves performance comparable to SFT on six math datasets, while significantly reducing the average number of generated tokens by up to 50%, thereby effectively mitigating the issue of overthinking. Moreover, QFFT exhibits superior performance compared to SFT in noisy, out-of-domain, and low-resource scenarios.

## 2 Towards Adaptive Reasoning

### 2.1 Definition of Adaptive Reasoning

Chain-of-Thought (CoT), since its introduction, has been widely adopted in various models (e.g., Qwen2.5-32B-Instruct), significantly enhancing its reasoning capabilities. With recent advancements, modern large reasoning models (e.g., DeepSeek-R1) increasingly employ **Long CoT** reasoning patterns, characterized by reflective verification behaviors [19, 6]. Reasoning patterns exhibiting

Table 1: Comparison of Reasoning Modes: Short CoT, Long CoT, and Adaptive Reasoning

|  | Short CoT | Long CoT | Adaptive Reasoning |
|---|---|---|---|
| **Reasoning Style** | Direct, concise. | Reflective, self-correction, containing reflective keywords (e.g., "wait"). | Adaptively uses Short/Long CoT based on question difficulty |
| **Application Scenarios** | Effective for simple questions. | Strong on difficult questions. | Optimal for both simple and difficult questions. |
| **Cost** | Low (fewer tokens). | High (redundant tokens). | Moderate (adaptive to needs). |
| **Risk** | Oversimplification. | Overthinking. | Balanced (mitigates both risks). |

these reflective verification behaviors can be categorized as Long CoT [20], whereas traditional CoT reasoning patterns lacking such behaviors are classified as **Short CoT**.

Despite the effectiveness of Long CoT in difficult questions, Short CoT maintains efficiency advantages for simple questions. This creates a fundamental trade-off. As shown in Table 1, the reflective capabilities of Long CoT improve accuracy but induce "overthinking", resulting in approximately 74.8% redundant tokens on simple questions; conversely, the conciseness of Short CoT, while efficient, risks "oversimplification", leading to a 75.1% average accuracy decline in difficult questions (shown in Figure 1).

> ***Key Observation 1:*** Long CoT patterns significantly outperform Short CoT on difficult questions, whereas Short CoT patterns achieve comparable performance with substantially fewer tokens on simple ones.

Key Observation 1 motivates us to explore methods to develop adaptive models to flexibly use Long CoT and Short CoT reasoning patterns based on question difficulty. In this paper, we formally define this capability as **Adaptive Reasoning**:

**Definition 1** *Adaptive Reasoning refers to a model's ability to adaptively prioritize Short CoT reasoning patterns for simple questions and prioritize Long CoT reasoning patterns for difficult questions when Short CoT is ineffective.*

Here, *simple questions* are those solvable by the model using Short CoT patterns, while *difficult questions* are those for which Short CoT fails.

## 2.2 Quantitative Metric on Adaptive Reasoning Ability

The adaptive reasoning capability of a model can be evaluated by measuring the alignment between its reasoning patterns and question difficulty. Here, question difficulty is based on whether the Short CoT of the evaluated model can correctly answer the question.

**Assumption 1** *If a Long CoT model $M_L$ is derived (e.g. distilled) from a Short CoT model $M_S$, we approximate the Short CoT capability of model $M_L$ by that of model $M_S$.*

Consequently, we introduce the Short CoT model $M_S$ as a **reference model** to estimate whether the evaluated model can correctly answer the question using Short CoT reasoning.

We introduce the **Reasoning Adaptability Cohen's Kappa (RAK)**, inspired by Cohen's Kappa [21], a statistical measure evaluating the agreement between two raters beyond chance. In our context, the two "raters" correspond to the question difficulty provided by the reference model (simple or difficult) and the reasoning patterns (Short or Long CoT) used by the evaluated model.

**Definition 2** ***Reasoning Adaptability Cohen's Kappa (RAK).**** If the Long CoT model $M_L$ is derived (e.g., distilled) from the original Short CoT model $M_S$ (reference model), RAK measures the performance of model $M_L$ in adaptively selecting the appropriate reasoning pattern, accounting for

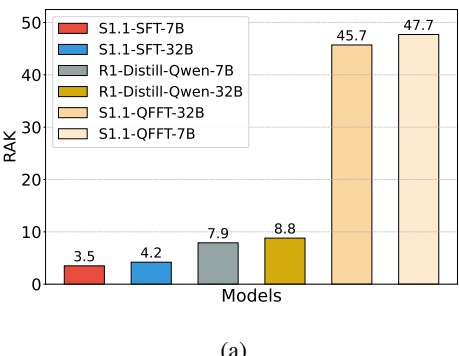 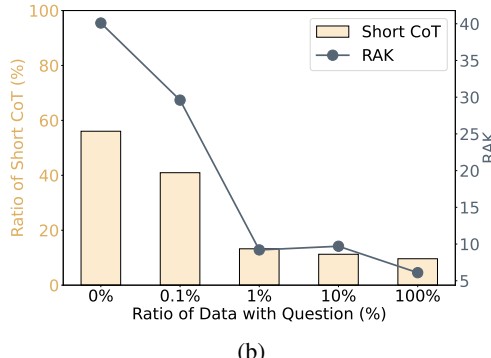

(a)                    (b)

Figure 2: Subfigure (a) presents the Reasoning Adaptability Cohen's Kappa (RAK) scores of different Long CoT models. Subfigure (b) illustrates that during Long CoT SFT, the Q (question) → R (Long CoT) mapping leads to Long CoT patterns overriding the model's original Short CoT patterns.

chance agreement.

$$\text{RAK} = \frac{p_o - p_e}{1 - p_e}, \tag{1}$$

where the observed agreement $p_o = \frac{\text{TP+TN}}{\text{TP+FP+FN+TN}}$, and the expected agreement $p_e = \frac{(\text{TP+FP})(\text{TP+FN}) + (\text{FN+TN})(\text{FP+TN})}{(\text{TP+FP+FN+TN})^2}$, computed based on the marginal probabilities of the predicted and actual classes [3].

A higher RAK indicates greater consistency between the evaluated model's reasoning pattern selection and task difficulty, demonstrating stronger adaptive reasoning capability. Conversely, a lower RAK reflects weaker consistency, indicating poorer adaptive reasoning capability.

## 2.3 Pilot Study on Adaptive Reasoning

We evaluate the reasoning adaptability of existing Long CoT models on the MATH500 [22] dataset. As shown in Figure 2a, we observe that current Long CoT models (obtained by SFT) have low RAK scores, which attributes to their consistent over-reliance on Long CoT reasoning patterns.

**Hypothesis.** In Long CoT SFT, models are trained on large-scale *(Q: question, R: Long CoT response)* pairs, teaching them to solve questions with Long CoT reasoning patterns. This training paradigm poses a potential risk [18]: models may default to Long CoT reasoning for all inputs, a phenomenon we term *Override*. We hypothesize that this phenomenon occurs because the new $Q \rightarrow R$ mapping—from questions to Long CoT responses—overwrites the original mapping to Short CoT responses, causing the model to over-rely on Long CoT patterns.

To verify this hypothesis, we conduct the following experiment: during Long CoT SFT, we randomly select a proportion $\alpha$ $(0 < \alpha < 1)$ of training samples and retain their original question-response pairs. For the remaining $(1 - \alpha)$ proportion of samples, we remove the questions, keeping only the Long CoT responses. We then evaluate how the ratio of the model's Short CoT changes as we gradually increase the value of $\alpha$.

**Results.** As shown in Figure 2b, when questions are removed from all training samples, the model exhibits over 50% Short CoT patterns[4] and does not over-rely on Long CoT reasoning. However, once samples with questions are included, even with a minimal logarithmic-scale increase (from 0.1% to 1%), the proportion of Short CoT patterns drops dramatically (from 40.95% to 13.24%). This

---

[3]Specifically, TP is the number of solvable questions of $M_S$ where model $M_L$ selects Short CoT patterns, FN is the number of solvable questions of $M_S$ where $M_L$ selects Long CoT patterns, FP is the number of unsolvable questions of $M_S$ where model $M_L$ selects Short CoT patterns, and TN is the number of unsolvable questions of $M_S$ where $M_L$ selects Long CoT patterns. Here, "solvable" indicates that the reference model $M_S$ answers the question correctly.

[4]We will discuss in Section 4 why a significant number of Long CoT patterns still appear.

indicates that even an extremely small proportion of $Q \to R$ mappings is sufficient to override the model's original Short CoT patterns, leading to an over-reliance on Long CoT reasoning.

> **Key Observation 2:** Current Long CoT models over-rely on Long CoT patterns. This is because the questions $(Q)$ to Long CoT responses $(R)$ mapping in SFT stage overrides their original Short CoT patterns.

# 3    Methodology: Question-Free Fine-Tuning

## 3.1    Motivation

To achieve adaptive reasoning, we aim for the model to (1) default to Short CoT patterns, while (2) adaptively triggering the reflective Long CoT behaviors when errors or uncertainty occur.

***First***, to preserve the model's default Short CoT patterns and prevent it from being overridden, we need to avoid training the model to learn a fixed $Q \to R$ mapping.

***Second***, the model should effectively learn Long CoT reasoning patterns, which could trigger reflective behaviors when the model is uncertain about its solution or encounters mistakes. Prior studies [6, 23] suggest that the core of Long CoT patterns lies in the structure of responses, rather than in the questions. This implies that models distilled solely from Long CoT responses, even without access to the corresponding questions, can still acquire Long CoT reasoning capability.

## 3.2    The Method

To preserve the original Short CoT patterns while enabling adaptive switching to reflective Long CoT patterns for challenging questions during distillation, we propose Question-Free Fine-Tuning (QFFT). Different from SFT, we remove the input question $Q$ entirely, training the model exclusively on the reasoning response $R$. Specifically, the model is optimized using a standard causal language modeling objective over the reasoning sequence:

$$\mathcal{L}_{\text{QFFT}} = -\frac{1}{|\mathcal{R}|} \sum_{t \in \mathcal{R}} \log P_\theta(R_t \mid R_{<t}, \cancel{\varnothing}), \tag{2}$$

where $\mathcal{R}$ is the set of token indices in the response sequence $R$ and $P_\theta$ is the model's output probability. $\cancel{\varnothing}$ indicates that QFFT removes the question component compared to SFT.

By omitting the question $Q$ during training and training solely on the reasoning response $R$, QFFT avoids learning a fixed mapping from questions to Long CoT responses while still acquiring Long CoT reasoning capability.

## 3.3    Why QFFT Leads to Adaptive Reasoning?

We explain why QFFT achieves adaptive reasoning from both training and inference stages.

**Training: Preserving Short CoT and Learning Reflection.**    During the QFFT training, the model avoids learning a direct mapping from questions (Q) to Long CoT responses (R), thereby preserving the model's default Short CoT patterns to answer questions. Additionally, the model is trained exclusively on Long CoT responses. Thus, the model theoretically acquires the capability for Long CoT reasoning and learns to exhibit reflective behaviors when encountering uncertainty or errors within the context of Long CoT reasoning. Formally, let $U_L$ denote the event of encountering uncertainty or errors during Long CoT reasoning. Then, the model learns the conditional probability:

$$P_\theta(B_r \mid U_L),$$

where $B_r$ denotes the occurrence of reflective behaviors, and $U_L$ represents uncertainty or errors arising specifically within the context of Long CoT reasoning.

**Inference: Default to Short CoT with Adaptive Reflection.**   At inference time, the QFFT model defaults to the Short CoT patterns. However, since the model has only explicitly learned the conditional probability $P_\theta(B_r \mid U_L)$ during training, it is not immediately obvious why it can still trigger reflective behaviors in the context of Short CoT reasoning. We explain this phenomenon from a transfer learning [24, 25] perspective:

**Assumption 2** *If a model has learned the conditional probability $P_\theta(B_r \mid U_L)$, this reflective capability can be transferred to Short CoT, enabling it to implicitly learn $P_\theta(B_r \mid U_S)$, where $U_S$ denotes uncertainty or errors that arise specifically within Short CoT reasoning.*

We empirically verify this assumption with experiments in Appendix F.5. Consequently, when the model detects errors or uncertainty in Short CoT reasoning context, it naturally triggers reflective behaviors to reconsider and correct its reasoning process.

In summary, QFFT enables the model to reason in an adaptive manner: it defaults to efficient Short CoT patterns and adaptively uses reflective Long CoT patterns when necessary, thus achieving both effectiveness and efficiency.

## 4  Experiments

In this section, we conduct experiments on both in-domain and out-of-domain datasets to validate the effectiveness of QFFT. Additional results (e.g. on different backbones) can be found in the appendix.

Table 2: Main results on 3 mathematical benchmarks. The reported accuracy and Avg. Len (Tokens) is averaged over 16 random sampling runs. RAK is our defined reasoning adaptability metric.

| Data | Method | GSM8K | | | MATH | | | AIME25 | | | Average | | |
|---|---|---|---|---|---|---|---|---|---|---|---|---|---|
| | | Acc ↑ | Tokens ↓ | RAK ↑ | Acc | Tokens | RAK | Acc | Tokens | RAK | Acc | Tokens | RAK |
| | | *7B Models (Based on Qwen2.5-7B-Instruct)* | | | | | | | | | | | |
| | SFT | 90.6 | 1.7K | 1.8 | 80.8 | 5.3K | 3.5 | 18.2 | 17.7K | 0 | 63.2 | 8.2K | 1.8 |
| S1.1 | QFFT | 91.0 | 0.4K | 28.4 | 80.2 | 2.8K | 47.7 | 17.2 | 12.8K | 28.0 | 62.8 | 5.3K | 34.7 |
| | Δ | +0.4 | -76.5% | +26.6 | -0.6 | -47.2% | +44.2 | -1.0 | -27.7% | +28.0 | -0.4 | -50.5% | +32.9 |
| | SFT | 88.2 | 1.8K | 0.2 | 80.4 | 5.8K | 6.1 | 16.8 | 17.1K | 0.2 | 61.8 | 8.2K | 2.2 |
| LIMO | QFFT | 88.0 | 0.7K | 26.7 | 80.6 | 4.1K | 40.1 | 17.2 | 15.6K | 34.2 | 61.9 | 6.8K | 33.7 |
| | Δ | -0.2 | -61.1% | +26.5 | +0.2 | -29.3% | +34.0 | +0.4 | -8.8% | +34.0 | +0.1 | -33.1% | +31.5 |
| | SFT | 91.0 | 1.4K | 2.3 | 81.6 | 5.7K | 4.6 | 19.8 | 13.8K | 0.1 | 64.1 | 7.0K | 2.3 |
| BS-17K | QFFT | 90.4 | 0.4K | 29.7 | 81.4 | 2.2K | 44.6 | 18.3 | 9.7K | 29.7 | 63.4 | 4.1K | 34.7 |
| | Δ | -0.6 | -71.4% | +27.4 | -0.2 | -61.4% | +40.0 | -1.5 | -29.7% | +29.6 | -0.8 | -54.2% | +32.3 |
| | | *32B Models (Based on Qwen2.5-32B-Instruct)* | | | | | | | | | | | |
| | SFT | 92.8 | 2.1K | 0.2 | 93.1 | 4.1K | 4.2 | 48.6 | 16.2K | 0 | 78.2 | 7.5K | 1.5 |
| S1.1 | QFFT | 93.6 | 0.6K | 31.2 | 92.2 | 2.4K | 45.7 | 46.8 | 12.9K | 29.3 | 77.5 | 5.3K | 35.4 |
| | Δ | +0.8 | -71.4% | +31.0 | -0.9 | -41.5% | +41.5 | -1.8 | -20.4% | +29.3 | -0.6 | -44.4% | +33.9 |
| | SFT | 91.2 | 1.9K | 1.0 | 93.0 | 3.9K | 8.8 | 45.8 | 13.2K | 0 | 76.6 | 6.3K | 3.3 |
| LIMO | QFFT | 92.6 | 0.8K | 27.4 | 92.6 | 2.9K | 38.9 | 45.0 | 12.5K | 33.2 | 76.7 | 5.4K | 33.2 |
| | Δ | +1.4 | -57.9% | +26.4 | -0.4 | -25.6% | +30.1 | -0.8 | -5.3% | +33.2 | +0.1 | -29.6% | +29.9 |

### 4.1  Experimental Setup

**Dataset.**   We select several high-quality distillation datasets, including S1.1 (1k) [9], LIMO (871) [10], and Bespoke-Stratos (17k) [8], where all responses are distilled from the DeepSeek-R1 model. Detailed data description is shown in Appendix E.1.

**Training details.**   We use Qwen2.5-Instruct-7B and Qwen2.5-Instruct-32B [26] as the base models. All experiments are conducted with LLaMA Factory [27], with a maximum sequence length of 16,384 tokens. Detailed hyperparameters can be found in the Appendix E.2.

**Evaluation.**   We evaluate model performance on six in-domain math datasets, including two simple datasets: **GSM8K** [28] and **MATH500** [22], two medium-difficulty datasets: the American Mathematics Competitions (**AMC23**) and **Minerva** [29], which includes undergraduate-level STEM

Table 3: Comparison with other Long-to-Short baselines. AES is used as a metric that balances performance and response length.

| Method | GSM8K | | | MATH | | | AIME25 | | | Average. | | |
|---|---|---|---|---|---|---|---|---|---|---|---|---|
| | Acc ↑ | Tokens ↓ | AES ↑ | Acc ↑ | Tokens ↓ | AES ↑ | Acc ↑ | Tokens ↓ | AES ↑ | Acc ↑ | Tokens ↓ | AES ↑ |
| *Long-to-Short Methods (7B)* | | | | | | | | | | | | |
| LIMO 7B (base) | 88.2 | 1.8K | - | 80.4 | 5.9K | - | 16.8 | 17.8K | - | 61.8 | 8.5K | - |
| SFT Shortest | 88.9 | 1.2K | 3.4 | 78.3 | 4.8K | -0.7 | **17.9** | 17.3K | 0.9 | 61.7 | 7.8K | 1.2 |
| DPO Shortest | 89.8 | 1.6K | 1.5 | 79.5 | 5.4K | -0.1 | 17.3 | 17.1K | 0.7 | 62.2 | 8.0K | 0.7 |
| SimPO Shortest | 87.2 | 1.2K | 2.2 | 75.8 | **3.2K** | -1.0 | 14.0 | **8.8K** | -12.0 | 59.0 | **4.4K** | -3.6 |
| O1-pruner | **90.8** | 0.8K | 5.8 | 78.2 | 3.2K | 1.8 | 14.2 | 12.2K | -12.7 | 61.0 | 5.4K | -1.7 |
| *Distilled Methods (7B)* | | | | | | | | | | | | |
| DAD-7B | 90.0 | 0.9K | 5.2 | 80.2 | 4.8K | 1.6 | 17.3 | 17.7K | 0.3 | **62.5** | 7.8K | 2.4 |
| QFFT (Ours) | 88.0 | **0.7K** | **5.9** | 80.6 | 4.1K | **2.9** | 17.2 | 15.6K | **1.4** | 61.9 | 6.9K | **3.4** |
| *Long-to-Short Methods (32B)* | | | | | | | | | | | | |
| LIMO 32B (base) | 91.2 | 1.9K | - | 93.0 | 3.9K | - | 45.8 | 13.2K | - | 76.7 | 6.3K | - |
| SFT Shortest | 93.2 | 9.2K | -38.2 | 91.4 | 3.1K | 0.4 | **46.3** | 14.5K | -0.9 | **76.9** | 8.9K | -12.9 |
| SimPO shortest | 94.5 | **0.6K** | 7.2 | 89.8 | 2.4K | 0.4 | 40.0 | 13.0K | -12.5 | 74.8 | 6.7K | -1.6 |
| O1-pruner | **94.8** | **0.6K** | 7.2 | 90.5 | **2.0K** | 2.1 | 29.2 | **7.3K** | -136.9 | 71.5 | **3.3K** | -42.5 |
| *Distilled Methods (32B)* | | | | | | | | | | | | |
| QFFT (Ours) | 92.6 | 0.8K | 5.9 | **92.6** | 2.9K | **2.1** | 45.0 | 12.5K | **-1.2** | 76.7 | 5.4K | **2.3** |

problems, and two high-difficulty datasets: **AIME 2024** and **AIME 2025**. Additionally, we also evaluate on two out-of-domain non-math datasets (in Section B.2): **GPQA** [30] and **MMLU-Pro** [31]. We report average accuracy by generating 16 responses per question using a sampling temperature of 0.6 and a maximum decoding length of 32,768 tokens. For calculating the Reasoning Adaptability score, we average the RAK scores of 16 samples to mitigate random fluctuations. Specifically, for the 7B SFT and QFFT models, we use Qwen2.5-7B-Instruct as the reference model, while for the 32B models, we use Qwen2.5-32B-Instruct. Further details regarding prompts and inference settings are provided in Appendix E.3.

## 4.2 Baselines

**Difficulty-Adaptive Distillation.** To enable adaptive reasoning, a straightforward approach is Difficulty-Adaptive Distillation (DAD), which combines short and Long CoT data for SFT based on question difficulty. Specifically, we use GSM8K as simple questions and LIMO as challenging questions, distilling Short-CoT responses for the former and Long-CoT responses for the latter. The combined data is then used SFT. Further details are provided in Appendix E.4.

**Long-to-Short Baselines.** We compare with four competitive *Long-to-Short* baselines: (1) **SFT-Shortest** [11]: SFT on the shortest correct responses directly. (2) **DPO-Shortest** & **SimPO-Shortest** [11]: widely used baselines in reasoning optimization. Following the setup of the original work, the shortest correct response is selected as the preferred sample, and the longest correct response as the rejected one. (3) **O1-Pruner** [14]: a reinforcement learning-based method that reduces reasoning length while preserving accuracy. It first builds a baseline via pre-sampling, then guides the model to generate more concise reasoning under performance-preserving constraints. We reproduce their results based on the LIMO model, following the experimental setup and hyperparameters as described in their paper.

**Efficiency Metric.** Following [14], we adopt the Accuracy–Efficiency Score (AES) to quantify the trade-off between model accuracy and token reduction. Let $(A_{\text{base}}, L_{\text{base}})$ and $(A_{\text{model}}, L_{\text{model}})$ denote the accuracy and token count for the baseline and evaluated models, respectively. In the Long-to-Short scenario, the baseline is the original Long CoT model (e.g., LIMO-7B); for QFFT, it is the model fine-tuned via SFT on the same dataset.

We define relative changes as: $\Delta L = \frac{L_{\text{base}} - L_{\text{model}}}{L_{\text{base}}}$, $\Delta A = \frac{A_{\text{model}} - A_{\text{base}}}{A_{\text{base}}}$. The AES is then computed as:

$$\text{AES} = \begin{cases} \alpha \, \Delta L + \beta \, |\Delta A|, & \Delta A \geq 0, \\ \alpha \, \Delta L - \gamma \, |\Delta A|, & \Delta A < 0, \end{cases}$$

with default parameters $\alpha = 0.1$, $\beta = 0.1$, and $\gamma = 1.0$. The AES is calculated by weighting and summing the model's solution token length and accuracy. In this metric, we prioritize maintaining accuracy (i.e., avoiding performance degradation) over reducing token usage.

### 4.3 Main Results

#### 4.3.1 Comparison with SFT

As shown in Table 2, we observe that across three mathematical evaluation datasets, QFFT methods significantly reduce the average token length while achieving performance comparable to that of SFT. In addition, QFFT substantially increases the **RAK** score, indicating improved adaptive reasoning capabilities. This suggests that QFFT effectively leverages Short CoT patterns when handling simple and solvable problems, reducing token consumption without sacrificing accuracy. A more detailed analysis of performance and reasoning adaptability is provided in the Section 5.

We further observe that the degree of token reduction varies with dataset difficulty. Specifically, QFFT achieves more substantial token savings on simpler datasets such as GSM8K and MATH, where the model adaptively retains a higher proportion of the Short CoT patterns. In contrast, on the more challenging AIME25 dataset, the model must rely more heavily on the Long CoT patterns, resulting in a less pronounced reduction in token consumption.

Finally, QFFT demonstrates strong adaptability across all three training datasets, including S1.1, LIMO, and BS-17K, when applied to both 7B and 32B model scales. This highlights the robustness and scalability of the QFFT approach.

> **Key Observation 3:** QFFT achieves performance comparable to SFT, significantly improving Reasoning Adaptability (RAK), thus substantially reducing the required token budgets.

#### 4.3.2 Comparison with Long-to-Short Baselines

As shown in Table 3, compared to Long-to-Short methods like SFT-Shortest and DPO, QFFT achieves greater token reduction with less performance degradation. On the other hand, while O1-Pruner and Simpo-FCS result in higher token reduction, they come at the cost of a significant drop in performance. In contrast, our QFFT method achieves higher AES values, providing a better trade-off between performance and token efficiency.

## 5 An In-depth Analysis

In this section, we conduct an in-depth analysis to address two key questions: **Q1:** *How QFFT Enables Adaptive Reasoning?* (Section 5.1) and **Q2:** *Why QFFT Matches SFT Performance?* (Section 5.2)

### 5.1 How QFFT Enables Adaptive Reasoning? (Q1)

In Section 3.3, we have explained why QFFT enables adaptive reasoning. In this part, we aim to explore, in the inference stage, how QFFT models utilize Long CoT patterns. To this end, we first present a detailed case study illustrating the model's transition from the default Short CoT to the Long CoT reasoning pattern. Subsequently, we analyze the primary scenarios and conditions under which the model adaptively adopts Long CoT reasoning, providing deeper insights into QFFT's adaptive reasoning capabilities.

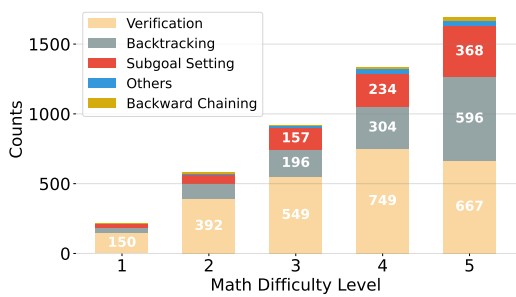

Figure 4: Long CoT Behavior Classification Trend on MATH500. The proportion of Long CoT increases by difficulty level.

**Case Study.** We check some cases of the reasoning chains of QFFT. As illustrated in Fig-

| **Question** |
| :--- |
| **Q:** Every morning Aya goes for a 9-kilometer-long walk and stops at a coffee shop afterwards... Find the number of minutes the walk takes her, including the t minutes spent in the coffee shop. |

| **QFFT  Long CoT Response** |
| :--- |
| To solve the problem, we need to determine the time Aya spends walking and in the coffee shop for different speeds. .. **Short** |
| ### Step 1: Convert all time units to hours. Since speed is in kilometers per hour... **CoT** |
| ### Step 2: First scenario: Walking at $s$ km/h ... Total time for the walk including coffee shop: 4 hours ... |
| ### Step 3: Second scenario: Walking at $s + 2$ km/h ... Here, a = 8, b = 11, c = -90: s = (-11 ± sqrt(11^2 - 4 * 8 * (-90)  * 5)) |
| / (2 * 8) s = (-11 ± sqrt(121 + 2880)) / 16 s = (-11 ± sqrt(2991)) / 16.   Check if sqrt (2991) is an integer. 2991 = 9 * 331, |
| so sqrt(2991) = 3 * sqrt(331), which is not an integer. This suggests a calculation error. **Wait**, let's recheck the quadratic |
| equation setup. **Wait,** after moving terms ... **Wait,** perhaps solving incorrectly. Let's compute the discriminant ... Therefore, no |
| real solution? **Wait,** check equations.  **Wait,** original equations ... Walking time at 3.0 km/h is 9 / 3.0 = 3 hours = 180 minutes. |
| Total time including t is 180 minutes + 24 minutes = 204 minutes. Thus, the answer is boxed{**204**}.   **Long  CoT** |

Figure 3:  A case study. This case illustrates that QFFT initially adopts the Short CoT patterns. Upon encountering an error, the model triggers the reflective Long CoT to refine its reasoning.

ure 3, we observe that in responses of the QFFT
model, **the reflective Long CoT patterns initially follow a concise Short CoT style** as expected.
Then, once the model encounters uncertainty or detects a potential error, it triggers Long CoT
reasoning patterns, marked by frequent use of reflective keywords such as "wait".

**Conditions Triggering Long CoT Reasoning.**   We further investigate under what conditions the
QFFT model would switch to Long CoT behavior.  According to [32], there are four main Long
CoT behaviors:  (1) **Verification**, verification or the systematic checking of intermediate results.
(2) **Backtracking**, which involves explicit revision of approaches when errors are detected.  (3)
**Sub-goal Setting**, where a complex problem is broken down into manageable steps. (4) **Backward
Chaining**, where in a goal-directed reasoning task, the solution is derived by working backward
from the desired outcome. Notably, adopting verification and backtracking behaviors suggests the
model faces uncertainties or errors. The four behaviors identified in our case study are illustrated in
Appendix F.4.

Then, we check responses generated by the QFFT model and observe that most Long CoT responses
are triggered by uncertainties or errors. Specifically, we use GPT-4o to classify the triggered Long
CoT behaviors in MATH into the four categories.  We regard the first "wait" as the boundary
between the two patterns, providing GPT-4o with two sentences before and after the first "wait" for
classification. As shown in Figure 4, verification is the most common one across all difficulty levels,
accounting for approximately 53% on average in Long CoT behaviors.  This indicates the QFFT
model tends to trigger Long CoT patterns when it is uncertain about the previous steps. The second
most common trigger is backtracking, which accounts for 26% on average, that is, rechecking and
updating previously incorrect steps.  Interestingly, as the difficulty of the questions increases, the
proportion of backtracking gradually increases, suggesting that the model reflects and updates its step
more frequently in such questions.

**Conclusion.**   Our experiments above illustrate Long CoT reasoning in QFFT is mostly triggered
by errors or uncertainties from the initial Short CoT patterns. The QFFT model defaults Short CoT
reasoning and dynamically activates Long CoT patterns when these uncertainties and errors arise,
demonstrating the reasoning adaptability.

## 5.2   Why QFFT Matches SFT Performance? (Q2)

Our experimental results indicate that the QFFT model achieves performance comparable to SFT
distillation primarily due to two key factors:

1. The model selectively employs Short CoT reasoning for simple problems, maintaining
   accuracy while significantly reducing token consumption.

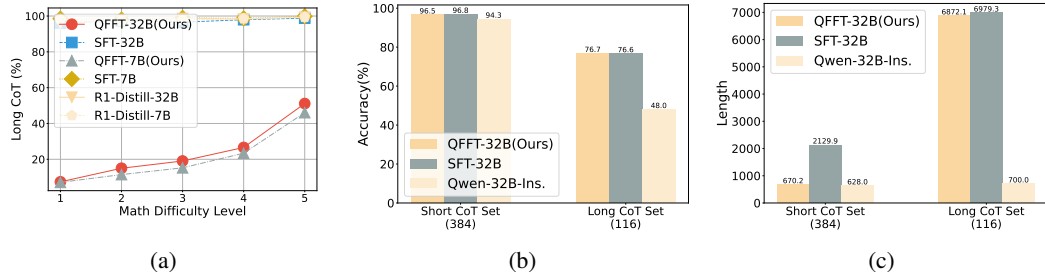

Figure 5: Subfigure (a) presents the proportion of Long CoT patterns used by different models across difficulty levels; Subfigures (b) and (c) evaluate the capabilities of Short CoT and Long CoT for QFFT, respectively, demonstrating that QFFT acquires Long CoT abilities comparable to SFT.

2. The model acquires the long-CoT reasoning capability comparable to that of SFT, enabling it to effectively handle challenging problems.

**QFFT Models Switch to Long CoT as Difficulty Rises.**  To investigate how the QFFT model adapts its reasoning patterns according to question difficulty, we analyze the proportion of Long CoT responses generated by the QFFT model on the MATH500 dataset. As illustrated in Figure 5a, the frequency of Long CoT responses clearly increases with problem difficulty—from just 7.41% on the easiest questions to 51.12% on the most challenging ones. This trend demonstrates that QFFT models effectively adopt an adaptive reasoning strategy, relying predominantly on Short CoT for simpler problems and increasingly invoking Long CoT reasoning as complexity grows.

**QFFT Acquires Long CoT Reasoning Capability Compared to SFT.**  To further evaluate the reasoning capabilities of the QFFT model under both short and Long CoT patterns, we conduct a detailed analysis using responses from the S1.1-QFFT-32B model on the MATH500 dataset. Specifically, we define two subsets: $\texttt{MATH}_{\text{System 1}}$, consisting of questions answered using Short CoT reasoning, and $\texttt{MATH}_{\text{System 2}}$, consisting of questions that trigger Long CoT reasoning. We then compare the performance of S1.1-QFFT-32B against Qwen2.5-32B-Instruct and S1.1-SFT-32B on these subsets.

As shown in Figures 5b and 5c, the S1.1-QFFT-32B demonstrates slightly improved Short CoT performance compared to the original Short CoT model, while its Long CoT capability remains comparable to the SFT model. Consequently, QFFT achieves overall performance on par with SFT distillation, while offering enhanced computational efficiency through adaptive reasoning.

# 6 Conclusion

In this paper, we revisit the complementary reasoning patterns of Short CoT and Long CoT models, and introduce Question-Free Fine-Tuning to effectively balance these two patterns. By removing explicit queries during training, QFFT preserves concise reasoning patterns from Short CoT while selectively leveraging the reflective reasoning capabilities of Long CoT. Experimental results on mathematical datasets demonstrate that our approach achieves performance comparable to SFT, while reducing the number of generated tokens by approximately 50%. Furthermore, QFFT exhibits superior generalization capabilities in out-of-domain tasks, noisy data , and low-resource scenarios.

# 7 Acknowledgement

This work was supported by Major Frontier Exploration Program (Grant No. C10120250085) from the Shenzhen Medical Academy of Research and Translation (SMART), the Shenzhen Science and Technology Program (JCYJ20220818103001002), NSFC grant 72495131, Shenzhen Doctoral Startup Funding (RCBS20221008093330065), Tianyuan Fund for Mathematics of National Natural Science Foundation of China (NSFC) (12326608), Shenzhen Science and Technology Program (Shenzhen Key Laboratory Grant No. ZDSYS20230626091302006), and Shenzhen Stability Science Program 2023.

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

# A  Two Equivalence of the QFFT Method

To understand (1) how QFFT preserves the model's original Short CoT capability and prevents them from being overridden and, (2) how QFFT enhances the model's Long CoT capabilities, we examine two theoretical equivalences of our method.

**Supervised Fine-tuning with Null Questions.**   QFFT can be conceptualized as a specialized form of SFT with null questions. In conventional SFT, models learn to map from questions $Q$ to Long CoT responses $R$, which overrides the original Short CoT. In contrast, with QFFT, since the questions are empty, the model does not learn any concrete $Q \to R$ mappings. Consequently, during inference, any non-empty input will not trigger Long CoT patterns, thereby preserving the model's original Short CoT patterns. In practice, this "null-question SFT" approach is both elegant and efficient, requiring no complex architectural modifications or specialized training strategies; it can be implemented simply by removing the question from the standard SFT frameworks.

**Continued Pre-training.**   From another perspective, QFFT functions as a specialized form of continued pre-training. Among training stages of LLMs, continued pre-training has emerged as an essential approach for expanding foundational capabilities and developing domain expertise [33, 34]. Unlike SFT, continued pre-training does not necessarily incorporate complete question-answer pairs and typically operates without question masking, incorporating loss calculations across all tokens. Consequently, QFFT can be conceptualized as a specialized form of continued pre-training that systematically enhances the model's general Long CoT proficiency, including reflective reasoning capabilities, by training exclusively on reasoning responses, without reliance on specific question-answer pair formats.

# B  More Applications of QFFT

Despite being trained without explicit question inputs, QFFT still consistently surpasses SFT in several targeted scenarios. In this section, we highlight three representative applications where QFFT demonstrates superior performance over SFT.

## B.1  Noisy Scenarios

In real-world applications, training data often contains various quality issues. In the context of Long CoT distillation, these issues may include incomplete reasoning processes, incorrect conclusions, or entirely irrelevant responses. Traditional SFT methods are highly sensitive to such noisy data, potentially leading to significant performance degradation.

To evaluate QFFT's robustness under various noise conditions, we design four progressively challenging noise levels as shown in Figure 6a:

**Level I: Normal Data**   As a baseline, this level uses original high-quality data containing complete and correct question-answer pairs.

**Level II: Incorrect Conclusions**   This level preserves the original reasoning process but introduces numerical errors in the final steps, leading to incorrect conclusions.

**Level III: Incomplete Reasoning**   This level randomly truncates approximately half of the answer content while maintaining the semantic completeness of sentences.

**Level IV: Irrelevant Answers**   This level creates completely mismatched question-answer pairs, where each question is paired with an answer from another question.

For each noise level, we construct datasets in both standard SFT and QFFT formats. We use the LIMO dataset as our distillation corpus, with Qwen2.5-7B-Instruct as the Short CoT model for distillation training. Performance is evaluated on MATH dataset by averaging results from 16 sampling runs per question.

Experimental results in Figure 6b demonstrate that as noise levels increase, standard SFT performance declines dramatically, from 76.5% at Level I to 0.4% at Level IV, indicating high sensitivity to noisy data. In the most severe case, with completely irrelevant responses (Level IV), the model completely loses its reasoning capabilities. In contrast, QFFT exhibits remarkable robustness, maintaining 78.6%

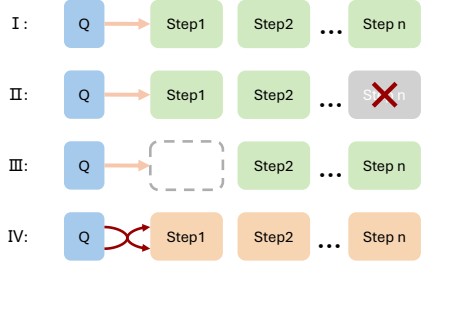
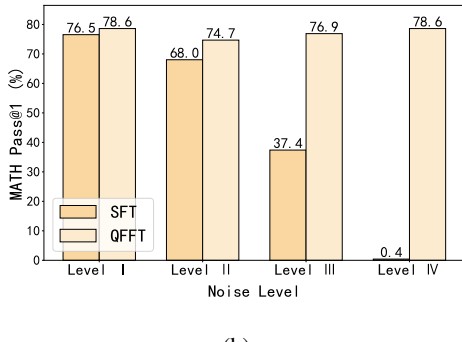

|  (a) | (b) |

Figure 6: Subfigure (a): Schematic illustration of different noise levels; Subfigure (b): Performance comparison between SFT and QFFT methods across different noise levels. As noise levels increase from Level I (normal data) to Level IV (irrelevant answers), SFT performance deteriorates dramatically, dropping from 76.5% to 0.4%. In contrast, QFFT maintains robust performance across all noise levels, achieving 78.6% even under extreme noise conditions such as Level IV.

performance even under extreme noise conditions at Level IV, matching its Level I performance. These results provide compelling evidence of QFFT's advantage in handling noisy training data.

> ***Key Observation 4:*** QFFT significantly outperforms SFT in handling noisy training data, maintaining robust performance even under extreme noise conditions where SFT methods completely fail.

## B.2 Out-of-Domain Scenarios

To assess the generalization ability on out-of-domain evaluation sets, we further conducted evaluations on the GPQA and MMLU-Pro datasets (detailed performance on specific subsets is provided in Appendix F.3). As shown in Table 4, (1) QFFT demonstrates superior performance compared to SFT on both **GPQA** [30] and **MMLU-Pro** [31] datasets, highlighting its strong generalization to out-of-domain data.

Additionally, to investigate whether the removal of queries by QFFT increases model **hallucinations**, we further evaluate the models on LLM-AggreFact[35], a benchmark specifically designed for hallucination detection. As shown in Table 4, models trained with SFT exhibit a noticeable decline in performance compared to the baseline Qwen2.5-Instruct models, especially for the 7B scale models. We observe that such decline is attributed to the models' poor instruction following ability, which leads to failures in extracting the final answers. In contrast, QFFT achieves slightly better results than the baseline on LLM-AggreFact, indicating that it does not exacerbate the risk of hallucinations.

> ***Key Observation 5:*** QFFT demonstrates superior out-of-domain generalization compared to SFT, and reduce hallucination risks.

## B.3 Low-Resource Scenarios

In certain low-resource scenarios (e.g., rare medical conditions [36] and unique geological phenomena [37]), high-quality data are often scarce, limiting the availability of extensive Long CoT training data. Under these conditions, Long CoT SFT typically struggles to enable models to effectively learn the Long CoT reasoning capability.

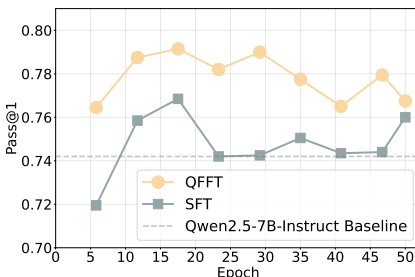

Figure 7: Performance of SFT and QFFT on MATH 500 in the low-resource scenario. The pass@1 score is averaged over 4 samples.

Table 4: Comparison on OOD dataset. LLM-AggreFact is used to evaluate **hallucination**. * means that SFT 7B models have poor instruction following ability and fail to extract most answers.

| Data | MMLU-Pro | GPQA | LLM-AggreFact |
|---|---|---|---|
| Qwen-2.5-Instruct-7B | 40.0 | 36.4 | 55.2 |
| + S1.1-SFT | 43.1 | 41.8 | 25.5* |
| + LIMO-SFT | 29.3 | 43.2 | 8.9* |
| + S1.1-QFFT | **53.2** | **44.4** | **58.9** |
| + LIMO-QFFT | 48.4 | 44.2 | 58.7 |
| Qwen-2.5-Instruct-32B | 62.2 | 49.5 | 75.3 |
| + S1.1-SFT | 64.9 | 60.6 | 60.5 |
| + LIMO-SFT | 52.4 | 65.3 | 62.2 |
| + S1.1-QFFT | **73.6** | 65.7 | **76.8** |
| + LIMO-QFFT | 73.4 | **67.9** | 75.8 |

To investigate the performance of QFFT in such scenarios, we simulate a low-resource scenario: We randomly sample 10 data points from the S1.1 dataset. To ensure training stability and sufficient data volume, each sampled data point is distilled 10 responses using Deepseek-R1, yielding a total of 100 training instances. Subsequently, we train both SFT and QFFT using these data with identical parameters for 50 epochs.

As depicted in Figure 7, QFFT consistently outperforms SFT in the low-resource scenario. Models trained with SFT predominantly rely on Long CoT patterns, which are not sufficiently internalized during training, resulting in limited overall performance. In contrast, QFFT not only retains the original Short CoT patterns but also adaptively uses Long CoT reasoning as needed. Consequently, QFFT achieves enhanced performance by integrating reflective Long CoT behaviors on top of the foundational Short CoT abilities, demonstrating superior performance in low-resource scenarios.

> *Key Observation 6:* QFFT demonstrates superior performance in low-resource scenarios compared to SFT.

## C  Related Work

**Large Reasoning Models**    Recent advancements in large reasoning models, such as OpenAI-o1 [1], DeepSeek-R1 [2], and QWQ [38], have significantly improved performance on complex tasks, such as mathematics and coding [3, 4, 5, 39]. These improvements largely stem from the test-time scaling paradigm, which enables models to adopt a Long CoT reasoning pattern—simulating human-like deep thinking through self-reflection, error correction, and multi-step solution exploration [39, 40, 41].

Building on this success, recent studies have explored distilling the capabilities of powerful Long CoT models into smaller, cost-effective ones. Some approaches leverage large-scale, diverse, and challenging datasets to equip smaller models with test-time scaling abilities. For example, works such as DeepSeek [2], OpenR1 [42], Sky-T1 [8], and OpenMathReasoning [43] curate broad, high-quality datasets and distill responses from DeepSeek-R1. These responses are used to fine-tune smaller dense models (e.g., Qwen2.5-32B) on reasoning patterns generated by the DeepSeek-R1 model, enabling them to outperform GPT-4 on mathematical reasoning benchmarks. On the other hand, methods such as S1 [9] and LIMO [10] fine-tune smaller models using only around 1000 high-quality and diverse examples paired with reasoning trajectories from DeepSeek-R1. Despite relying on a fraction of the data and computational resources, these approaches empower Short CoT models with strong reasoning capabilities, achieving competitive or even superior performance on challenging benchmarks.

**Long-to-Short Approaches**    While Long CoT has significantly improved the ability of LLMs to handle complex reasoning tasks, it often comes with substantial computational overhead and a tendency toward overthinking [11, 19]. To solve this problem, work [11] explores methods such as

DPO and SimPO, which sample multiple responses and train models to prefer the shortest and correct responses. O1-Pruner [14] proposes a Length Harmonizing Fine-Tuning framework that formulates an optimization objective to prune unnecessary steps while maintaining accuracy. Cot-Valve [16] dynamically controls the reasoning chain length, allowing the model to use shorter chains on simple tasks and retain longer ones for complex reasoning. On the other hand, [44] merges short and Long CoT models to achieve efficient reasoning. Other approaches [45, 46, 47, 48] further enhance the token efficiency of models by adopting approaches such as adaptively allocating reasoning budgets based on task difficulty. However, most existing methods focus primarily on improving the efficiency of Long CoT reasoning. This paper proposes a novel perspective: enabling models to adaptively learn to use Long CoT and Short CoT reasoning adaptively, achieving greater efficiency and flexibility in inference.

## D   Deeper Insights into the QFFT Method

The QFFT method provides a unique advantage in its ability to inject new patterns into a model without overriding the model's default patterns. Specifically, consider a scenario where a model initially possesses a default pattern (Pattern A, e.g., Short CoT). When additional distinct patterns (Patterns B and C, e.g., Long CoT) are introduced through training data, QFFT facilitates the seamless integration of these new patterns while preserving the original Pattern A. Consequently, the model can adaptively activate Patterns B and C based on their respective triggering conditions, achieving an adaptive utilization of multiple patterns.

Importantly, when Patterns A, B, and C each exhibit unique capabilities, meaning that no single pattern can fully replicate or substitute the functionality of another, the QFFT-trained model demonstrates clear superiority over traditional SFT. This is because SFT typically overwrites the default patterns (losing the capabilities of the default pattern A). In contrast, QFFT maintains the coexistence and adaptive triggering of multiple patterns, enabling the model to leverage the capabilities of diverse patterns.

Looking ahead, we plan to explore the injection of additional specialized patterns, such as tool-oriented patterns (e.g., API-calling patterns, code patterns) or even custom-designed patterns tailored explicitly to specific tasks. This direction promises further enhancement of model flexibility and adaptability, opening new avenues for advanced pattern integration and utilization.

## E   Experimental Details

### E.1   Training Data

In our experiments, both QFFT and SFT are trained using three primary datasets: S1.1, LIMO, and Bespoke-Stratos-17k. Below, we provide an overview of each dataset.

**S1.1**   [9]: The `S1.1` dataset consists of 1,000 carefully curated questions paired with detailed reasoning traces. These questions were selected based on three criteria: difficulty, diversity, and quality. The dataset covers a range of subjects, primarily focusing on mathematics, but also includes other scientific disciplines such as biology, physics, and economics.

**LIMO**   [10]: The `LIMO` dataset contains 817 high-quality training samples, each comprising a mathematical problem and its corresponding detailed reasoning chain. The reasoning chains were distilled from the DeepSeek-R1 model to ensure accuracy and clarity.

**Bespoke-Stratos-17k**   [8]: The `Bespoke-Stratos-17k` dataset includes 17,000 reasoning examples, each containing a question, a reasoning trace, and an answer. This dataset was generated by replicating and improving the Berkeley Sky-T1 data pipeline and utilizing SFT distillation data from DeepSeek-R1.

For all three datasets, the responses used in our experiments were based on the original responses from the datasets, which were distilled from the DeepSeek-R1 model. This ensures that the reasoning chains used in both training and evaluation are of high quality.

### E.2 Training Details

For the SFT baselines, we follow the official hyperparameters from S1, LIMO, and Sky-T1 during training. For QFFT, we use the same set of hyperparameters as those used for SFT. We use Llama Factory [27] for training. During the QFFT training, we modify the template in the Llama Factory by removing the question and all system prefixes, focusing solely on training the responses. Other detailed hyperparameters used in the training process are presented in Table 5.

Table 5: Hyperparameters for training different sizes of models in our experiments. Both SFT and QFFT share the same set of hyperparameters.

| Hyperparameter | 7B Models | 32B Models | Phi4-Mini-Instruct |
| --- | --- | --- | --- |
| Cutoff_len | 16384 | 16384 | 16384 |
| Batch_size | 8 | 32 | 8 |
| Learning_rate | 1e-5 | 1e-5 | 1e-5 |
| Epochs | 6 | 6 | 6 |
| Lr_scheduler_type | Cosine | Cosine | Cosine |
| Weight_decay | 1e-4 | 1e-4 | 1e-4 |
| Warmup_ratio | 0.1 | 0.1 | 0.1 |

---

**Evaluation Prompt**

**System:** Please reason step by step, and put your final answer within $\boxed{}$.
**User:** {Question}
**Assistant:**

---

Figure 8: The prompt of evaluation in our experiments.

### E.3 Evaluation Details

We conducted evaluation experiments on all models using $t = 0.6$ and a token budget of 32,768. For the mathematical datasets, we sample 16 times to compute the average accuracy. For the Out-of-Domain tasks (including GPQA, MMLU-Pro, and LLM-AggreFact), we run 4 times and report the average accuracy. We use the VLLM reasoning architecture, and the inference setup is aligned with LIMO. The specific prompts used are shown in Figure 8.

### E.4 Long-to-Short Baseline Details

**Difficulty-Adaptive Distillation**    To enable adaptive selection between long-CoT and short-CoT patterns, a straightforward approach is Difficulty-Adaptive Distillation (DAD) for short and long reasoning chains. Specifically, we identify the challenging subset $\mathcal{D}_{\text{hard}}$ (850 examples) by selecting questions from S1.1 that Qwen2.5-Instruct-7B fails to solve. For the simple subset $\mathcal{D}_{\text{easy}}$, we combine questions correctly answered by Qwen2.5-Instruct-7B from S1.1 and randomly sampled questions from the GSM8K training set, totaling 850 examples. For $\mathcal{D}_{\text{hard}}$, we use the solutions provided by S1.1, which are distilled from DeepSeek-R1. For $\mathcal{D}_{\text{easy}}$, we distill responses from the Qwen2.5-Instruct-72B model.

**Long-to-Short training**    Long-to-Short methods are typically employed to reduce the redundant reasoning lengths of long-CoT models, such as distillation models. In our experiments, we selected the LIMO-7B and 32B distillation models. We compared our approach against four competitive Long-to-Short baseline methods:

(1) **SFT-Shortest** [11]: For the LIMO dataset, we used LIMO distillation models to sample ten times. Following the procedure in [11], we constructed a dataset by selecting the shortest correct responses and then performed SFT.

(2) **DPO-Shortest** and **SimPO-Shortest** [11]: Adhering to the setup described in [11], we sampled ten times using the distilled LIMO model on the LIMO dataset. The shortest correct response was chosen as the preferred sample, while the longest correct response served as the rejected sample.

(3) **O1-Pruner** [14]: We replicated its results based on the LIMO distillation model. The experiments were conducted on the LIMO dataset, and all hyperparameters strictly followed the specifications outlined in the original paper.

# F    Additionally Experiments

## F.1    Experimental Results on More Datasets

In this section, we report the experimental results on AMC, AIME, and Minerva. Here we draw the same conclusion as in the main text. As shown in Table 6, (1) we observe that across three mathematical evaluation datasets, QFFT methods significantly reduce the average token length while achieving performance comparable to that of SFT. This can be attributed to the substantial improvement in reasoning adaptability brought about by QFFT, which leverages System 1 patterns for simple and solvable questions, thus reducing token consumption. (2) We observe that QFFT achieves a more significant token reduction on the simpler datasets (e.g., AMC) compared to AIME24. This is due to the relative simplicity of these datasets, which allows the model to retain a higher proportion of System 1 patterns, thus significantly reducing the average token length. In contrast, on the more challenging AIME24 dataset, QFFT retains fewer System 1 patterns, leading to a less pronounced token reduction.

Table 6: Main results on 4 mathematical benchmarks. The reported accuracy and Avg. Len (Tokens) are averaged over 16 random sampling runs.

| Data | Method | AMC | | | AIME24 | | | Minerva | | | Average | | |
|---|---|---|---|---|---|---|---|---|---|---|---|---|---|
| | | Acc | Tokens | RAK | Acc | Tokens | RAK | Acc | Tokens | RAK | Acc | Tokens | RAK |
| **7B Models (Based on Qwen2.5-7B-Instruct)** | | | | | | | | | | | | | |
| | SFT | 56.1 | 12.9K | 3.2 | 19.0 | 13.7K | 0.5 | 36.2 | 8.5K | 0.7 | 37.1 | 11.7K | 1.5 |
| S1.1 | QFFT | 55.3 | 7.4K | 43.7 | 20.6 | 10.3K | 29.6 | 38.2 | 2.9K | 27.8 | 38.0 | 6.9K | 33.7 |
| | Δ | -0.8 | -42.4 | +40.5 | +1.6 | -24.6 | +29.1 | +2.0 | -65.9 | +27.1 | +0.9 | -44.3 | +32.2 |
| | SFT | 55.8 | 12.6K | 0.7 | 19.1 | 13.9K | 0.0 | 38.2 | 7.3K | 1.1 | 37.7 | 11.3K | 0.6 |
| LIMO | QFFT | 57.2 | 9.8K | 39.8 | 19.6 | 11.2K | 28.9 | 37.7 | 4.6K | 24.2 | 38.2 | 8.5K | 31.0 |
| | Δ | +1.4 | -21.9 | +39.1 | +0.5 | -19.6 | +28.9 | -0.5 | -37.5 | +23.1 | +0.5 | -26.3 | +30.4 |
| | SFT | 61.6 | 11.2K | 3.6 | 19.4 | 13.4K | 0.0 | 40.8 | 4.6K | 0.9 | 40.6 | 9.7K | 1.5 |
| BS-17K | QFFT | 61.6 | 7.2K | 40.1 | 20.6 | 9.0K | 27.7 | 39.3 | 3.4K | 26.8 | 40.5 | 6.5K | 31.5 |
| | Δ | 0.0 | -36.1 | +36.5 | +1.2 | -32.7 | +27.7 | -1.5 | -25.7 | +25.9 | -0.1 | -31.5 | +30.0 |
| **32B Models (Based on Qwen2.5-32B-Instruct)** | | | | | | | | | | | | | |
| | SFT | 87.2 | 8.9K | 0.8 | 56.9 | 12.2K | 0.0 | 48.3 | 4.9K | 1.1 | 64.1 | 8.7K | 0.6 |
| S1.1 | QFFT | 86.9 | 5.7K | 38.7 | 56.7 | 9.4K | 24.9 | 47.1 | 3.1K | 31.1 | 63.6 | 6.1K | 31.6 |
| | Δ | -0.3 | -35.9 | +37.9 | -0.2 | -23.0 | +24.9 | -1.2 | -37.2 | +30.0 | -0.6 | -32.0 | +30.9 |
| | SFT | 91.7 | 9.2K | 2.4 | 56.7 | 10.7K | 0.6 | 46.7 | 4.4K | 1.4 | 65.0 | 8.1K | 1.5 |
| LIMO | QFFT | 89.5 | 7.2K | 42.1 | 54.6 | 9.5K | 22.8 | 46.1 | 3.5K | 28.9 | 63.4 | 6.8K | 31.3 |
| | Δ | -2.2 | -21.5 | +39.7 | -2.1 | -10.7 | +22.2 | -0.6 | -20.4 | +27.5 | -1.6 | -17.5 | +29.8 |

## F.2    Analysis with Different Model Backbones

To investigate whether the QFFT method is limited to specific backbone architectures (e.g., Qwen), we evaluate QFFT using an alternative backbone, Phi4-mini-Instruct, on the LIMO and S1.1 datasets. As shown in Table 7, compared to SFT, QFFT significantly enhances the reasoning adaptability under the Phi architecture, while achieving comparable overall performance. These results are consistent with those on Qwen architecture, demonstrating that QFFT is not constrained to a specific backbone and generalizes well to other architectures.

## F.3    Detailed Analysis on Out-of-Domain Benchmarks

In this section, we further analyze the generalization capabilities of QFFT on various subtasks from the MMLU-Pro benchmark across diverse domains. As shown in Table 8, after SFT, performance significantly decreases in knowledge-intensive domains such as Law, History, and Economics compared to the baseline (Qwen2.5-7B-Instruct). In contrast, improvements are notable in Physics and Chemistry.

Table 7: Comparison of SFT and QFFT based on the Phi-4-Mini-Instruct base model.

| Data | Method | GSM8K | | | MATH | | | AIME25 | | | Average | | |
|---|---|---|---|---|---|---|---|---|---|---|---|---|---|
| | | Acc | Tokens | RAK | Acc | Tokens | RAK | Acc | Tokens | RAK | Acc | Tokens | RAK |
| | | | | | **Based on Phi-4-mini-Instruct (3.8B)** | | | | | | | | |
| S1.1 | SFT | 88.6 | 2.4K | 1.1 | 69.8 | 5.3K | 2.7 | 9.7 | 15.8K | 0 | 56.0 | 7.8K | 1.3 |
| | QFFT | 88.9 | 0.9K | 31.4 | 69.7 | 2.8K | 42.9 | 10.2 | 12.8K | 26.5 | 56.3 | 5.5K | 33.6 |
| | Δ | +0.3 | -62.5% | +30.3 | -0.1 | -47.2% | +40.2 | +0.5 | -19.0% | +26.5 | +0.2 | -42.9% | +32.3 |
| LIMO | SFT | 88.1 | 1.7K | 0.8 | 67.2 | 5.2K | 7.1 | 8.3 | 16.1K | 1.8 | 54.5 | 7.7K | 3.2 |
| | QFFT | 88.0 | 0.6K | 29.3 | 66.9 | 3.7K | 38.1 | 9.1 | 12.5K | 31.6 | 54.7 | 5.6K | 33.0 |
| | Δ | -0.1 | -64.7% | +28.5 | -0.3 | -28.8% | +31.0 | +0.8 | -22.4% | +29.8 | +0.1 | -38.6% | +29.8 |

We analyze and have the following explanation: Physics and Chemistry share substantial knowledge overlap with Mathematics, and solving problems in these fields often inherently requires mathematical reasoning skills. Therefore, training the model on mathematical problems naturally enhances its generalization to Physics and Chemistry tasks. This observation aligns with findings reported in the study [10].

In contrast, domains such as Law, History, and Economics are primarily knowledge-intensive, relying more heavily on factual recall and contextual understanding rather than structured reasoning. Since SFT is driven by queries during training, it tends to align the model closely with specific question formats. This overfitting may limit the model's flexibility in generalizing to tasks outside the training distribution, particularly in domains that require broad knowledge rather than reasoning. The reliance on task-specific queries may overly anchor the model to narrow cues, leading to catastrophic forgetting of general knowledge and ultimately degrading performance in unrelated domains.

QFFT mitigates these question-induced biases by learning reasoning patterns independent of specific input queries. By doing so, QFFT effectively reduces task-specific overfitting, preserving general knowledge and improving the model's ability to generalize across diverse domains.

Table 8: Accuracy (%) on Various Categories of the MMLP-Pro Dataset. The background red indicates performance **improvement**, and the green indicates performance **decline** compared to Qwen2.5-7B-Instruct. Results are averaged over four runs.

| Category | LIMO-7B-QFFT | LIMO-7B-SFT | Qwen2.5-7B-Instruct |
|---|---|---|---|
| Business | 42.3 | 54.8 | 24.3 |
| Law | 29.5 | 5.5 | 53.0 |
| Psychology | 54.0 | 9.3 | 53.5 |
| Chemistry | 29.8 | 42.8 | 11.8 |
| History | 50.3 | 12.0 | 55.0 |
| Other | 50.8 | 21.3 | 49.0 |
| Health | 52.5 | 16.5 | 51.5 |
| Economics | 58.0 | 27.8 | 57.5 |
| Physics | 48.8 | 52.0 | 20.3 |
| Computer Science | 41.8 | 33.0 | 39.5 |
| Philosophy | 43.5 | 26.5 | 49.0 |
| Engineering | 20.5 | 31.0 | 15.8 |
| **All** | 48.4 | 29.3 | 40.0 |

## F.4 Case Study

In this section, we present specific cases to illustrate the four distinct behaviors exhibited by QFFT with Long CoT, as shown in Figure 9.

The **Verification** behavior involves reflective processes such as double-checking and validating intermediate results. For example, the model uses "let me double-check", which explicitly verifies intermediate computations to ensure correctness.

The **Backtracking** behavior involves explicitly revising previous steps when errors or inconsistencies are detected. For instance, upon encountering an error such as "but 2 is already used", the model traces back to earlier steps to identify and rectify the mistake.

The **Sub-goal Setting** behavior decomposes complex problems into multiple manageable sub-steps or sub-goals, facilitating step-by-step reasoning and problem-solving.

The **Backward Chaining** behavior refers to goal-directed reasoning, where the solution is derived by reasoning backward from the desired outcome. For example, the model may reason: "Wait, actually, if we can express $S$ in terms of $T$, then..." to guide its reasoning process toward the goal.

| Verification | Backtracking |
|---|---|
| …… So, the 4th value is 29, and the 5th value is 29 Therefore, the median is (29 + 29)/2 = 29 Alternatively, since both the 4th and 5th values are 29, their average is 29 **Wait**, but **let me double-check** the sorted list to make sure I didn't miss anything Original values …… | …… Let's list all possible ways to factor 84 into distinct positive integers and compute their sums First, let's consider the prime factors: 2, 3, 7 Their product is 2 × 3 × 7 = 42, which is less than 84, so we need another factor **Wait**, but 2 × 3 × 7 = 42, and 42 × 2 = 84, **but 2 is already used** Alternatively, 42 × 2 = 84, but 2 is already in the set So, we need to combine some factors …… |

| Subgoal Setting | Backward Chaining |
|---|---|
| …… alternate interior angles are equal, and consecutive interior angles are supplementary In the diagram, there is a transversal from point Q to point T, and another transversal from point R to point T **Wait**, maybe the lines PQ and TR are transversals **Let me try to figure out the relationships between the angles First**, angle at P is x degrees, angle at T is 2x degrees …… | …… So, 2T + G = S Our goal is to find the number of treeks that equal one squig, which is S/T But since the problem asks for how many treeks equal the weight of one squig, we need to find S/T **Wait**, actually, **if we can express S in terms of T**, then S/T would be the number of treeks equivalent to one squig But perhaps the problem is asking for the number n such that …… |

Figure 9: Behavior Classification Examples.

### F.5 Generalization of Reflective Behaviors from Short CoT to Long CoT

To validate our second assumption—that reflective capabilities learned by Long CoT models can transfer to Short CoT contexts—we designed an experiment to examine whether models trigger reflective behaviors when errors occur during Short CoT reasoning.

**Experimental Design** We utilized the Process-Bench [49] dataset, which contains detailed step-by-step solutions to mathematical problems produced by Short CoT models, with each step annotated as either correct or incorrect. This dataset enabled precise control over error conditions in our experiments. We constructed two types of reasoning sequences: completely correct reasoning step sequences and sequences containing errors. We then prompted Long CoT models (Deepseek-R1-Qwen-Distill-7B) to continue writing from these sequences and analyzed the model's behavioral differences, particularly focusing on the emergence of reflective behaviors.

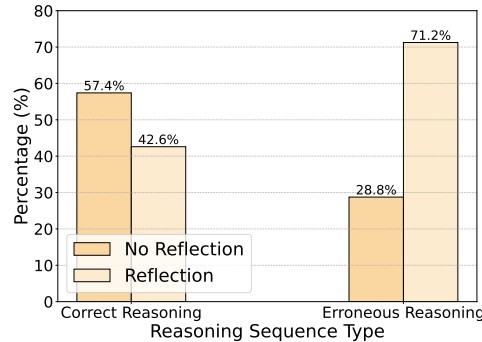

Figure 10: Reflection Behavior Comparison.

**Results** Our experimental results demonstrate that when Long CoT models are provided with correct reasoning steps, they tend to directly continue the reasoning process with minimal reflection as shown in Figure 10. Specifically, models exhibited reflective behaviors in approximately 42.6% of cases following correct reasoning sequences. In contrast, when presented with sequences containing errors, the proportion of reflective behaviors increased dramatically to 71.2%. These reflective

behaviors manifested as identifying calculation errors, revisiting reasoning paths, providing corrected solution approaches, and validating intermediate results.

These findings strongly support our hypothesis: the reflective capabilities of Long CoT models, characterized by $P_\theta(B_r \mid U_L)$, successfully transfer to Short CoT contexts, represented as $P_\theta(B_r \mid U_S)$.

# G   Limitations

The QFFT approach is simple yet efficient. Compared to other efficiency-oriented methods, it avoids complicated rejection sampling procedures and thus significantly saves computational resources. However, we observe that QFFT does not effectively optimize the efficiency of Long CoT reasoning. As a result, the overthinking issues still occur on challenging questions, such as AIME24 and AIME25. To address this limitation, we further explore combining QFFT with various Long-to-Short methods, aiming to effectively improve the efficiency of Long CoT reasoning within the QFFT framework.

Table 9: Comparison with other Long-to-Short baselines. AES is used as a metric that balances performance and response length.

| Method | GSM8K | | | MATH | | | AIME25 | | | AVG. | | |
|---|---|---|---|---|---|---|---|---|---|---|---|---|
| | Acc ↑ | Tokens ↓ | AES ↑ | Acc ↑ | Tokens ↓ | AES ↑ | Acc ↑ | Tokens ↓ | AES ↑ | Acc ↑ | Tokens ↓ | AES ↑ |
| *Long-to-Short Methods (7B)* | | | | | | | | | | | | |
| LIMO 7b (base) | 88.2 | 1.8K | - | 80.4 | 5.9K | - | 16.8 | 17.8K | - | 61.8 | 8.5K | - |
| SFT Shortest | 88.9 | 1.2K | 3.4 | 78.3 | 4.8K | -0.7 | **17.9** | 17.3K | 0.9 | 61.7 | 7.8K | 1.2 |
| DPO Shortest | 89.8 | 1.6K | 1.5 | 79.5 | 5.4K | -0.1 | 17.3 | 17.1K | 0.7 | 62.2 | 8.0K | 0.7 |
| SimPO Shortest | 87.2 | 1.2K | 2.2 | 75.8 | **3.2K** | -1.0 | 14.0 | **8.8K** | -12.0 | 59.0 | **4.4K** | -3.6 |
| O1-pruner | **90.8** | 0.8K | 5.8 | 78.2 | 3.2K | 1.8 | 14.2 | 12.2K | -12.7 | 61.0 | 5.4K | -1.7 |
| *Distilled Methods (7B)* | | | | | | | | | | | | |
| DAD-7B | 90.0 | 0.9K | 5.2 | 80.2 | 4.8K | 1.6 | 17.3 | 17.7K | 0.3 | **62.5** | 7.8K | 2.4 |
| QFFT (Ours) | 88.0 | 0.7K | 5.9 | 80.6 | 4.1K | 2.9 | 17.2 | 15.6K | 1.4 | 61.9 | 6.9K | 3.4 |
| + DPO Shortest | 86.4 | 0.7K | 4.0 | 80.5 | 4.0K | 3.2 | 16.7 | 15.2K | 0.5 | 61.2 | 6.6K | 2.6 |
| + SimPO Shortest | 89.0 | **0.5K** | **7.3** | 80.2 | **3.4K** | **4.1** | 17.7 | 13.6K | **2.9** | 62.3 | 5.8K | **4.8** |

**QFFT + Long-to-Short**   We investigate whether the QFFT-trained model can further benefit from the Long-to-Short strategy to enhance the efficiency of Long CoT reasoning. Specifically, we explore applying DPO-shortest and SimPO-shortest methods to the QFFT model. To achieve this, we first sample 10 responses from the LIMO-7B-QFFT model on the LIMO dataset and select the shortest correct solution among them for subsequent DPO and SimPO training. As shown in Table 9, we observe that both DPO and SimPO methods can further improve the overall token efficiency of the QFFT model. Notably, these methods significantly enhance the AES metric with minimal performance degradation.

We further compare the average length of Long CoT and Short CoT after applying DPO and SimPO. As shown in Figure 11, we find that the average token count of Long CoT decreases significantly. This indicates that the Long-to-Short method can be effectively applied to QFFT to further enhance the efficiency of Long CoT. This demonstrates the scalability and adaptability of the QFFT method, highlighting its potential to effectively integrate with Long-to-Short strategies for further efficiency gains.

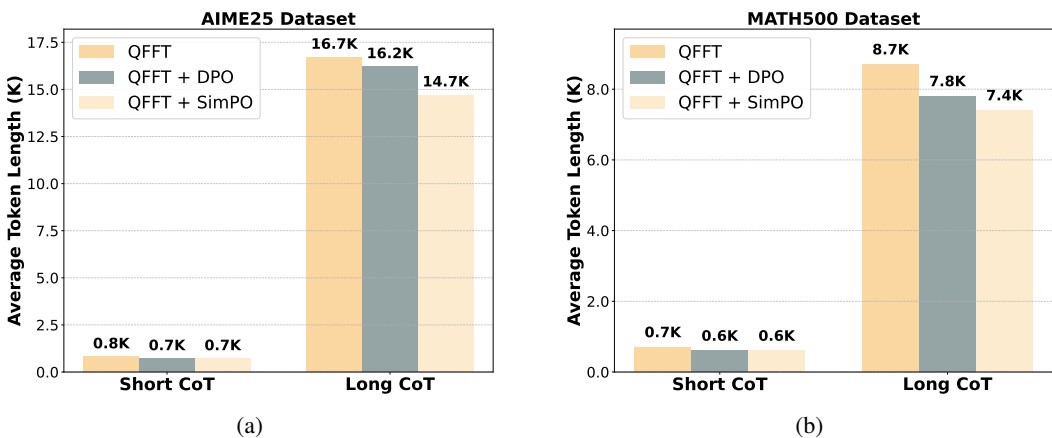

(a)                                                         (b)

Figure 11: Average token length comparison of LIMO-7B-QFFT models on mathematical problem-solving datasets. Subfigure (a) presents the AIME25 dataset and subfigure (b) presents the MATH500 dataset, showing the average lengths of Short CoT and Long CoT responses across three model variants: the base LIMO-7B-QFFT model, and the same model enhanced with either DPO or SimPO for length reduction (Long-to-Short).

