# OpenReview forum: "QFFT, Question-Free Fine-Tuning for Adaptive Reasoning"
_NeurIPS.cc/2025/Conference — NeurIPS 2025 spotlight_

### Official Review · Reviewer_CTuE · 2025-06-28

**Clarity:** 3
**Significance:** 2
**Originality:** 3
**Rating:** 5
**Confidence:** 4

**Summary:**

This paper introduces Question-Free Fine-Tuning (QFFT), a new method to improve the efficiency and adaptability of large language models. Instead of training with question–answer pairs, QFFT removes the input questions and fine-tunes only on reasoning chains. This allows the model to keep its original concise reasoning for simple tasks while selectively using longer, reflective reasoning for complex ones. Experiments show that QFFT reduces response length by over 50% without losing accuracy and generalizes better to new tasks. It effectively balances performance and token efficiency by enabling adaptive reasoning.

**Questions:**

1. As mentioned in the Weakness, could the authors consider including additional and stronger baselines?
2. How does the proposed method perform on other models, such as LLaMA? It would be helpful to include additional experiments to demonstrate the generalizability of the method across different model architectures.
3. Since the proposed fine-tuning method removes human user queries, it may potentially weaken the model’s instruction-following ability. Could the authors include experiments on instruction-following benchmarks to rule out this possibility?

**Ethical Concerns:**

["NO or VERY MINOR ethics concerns only"]

**Final Justification:**

The paper presents a well-structured study that introduces a simple yet effective method QFFT.  Through comprehensive in-domain and out-of-domain experiments, it demonstrates that QFFT maintains or slightly improves accuracy while significantly reducing token usage. Through the rebuttal, the authors effectively addressed my concerns regarding additional models and instruction-following performance, which led me to raise my score from 4 to 5.

**Limitations:**

See Weaknesses and Questions

**Quality:**

3

**Strengths And Weaknesses:**

**Strengths:**
1. The paper conducts comprehensive experiments on both in-domain (e.g., GSM8K, AIME) and out-of-domain (e.g., GPQA, MMLU-Pro) benchmarks, showing that QFFT maintains or slightly improves accuracy while significantly reducing token usage.
2. The paper clearly identifies the problem of overthinking in long CoT models and the inefficiency of existing distillation approaches. Besides, the paper is well-organized, with distinct sections for background, method, experiments, and analysis, each flowing logically from one to the next.
3. The core innovation—fine-tuning without the input question—is simple yet effective. It breaks from standard supervised fine-tuning and opens a new direction in adaptive reasoning.

**Weaknesses:**
1. To the best of my knowledge, there has been extensive research on effective reasoning. However, this work includes only a limited number of relatively weak baselines. Stronger and more diverse baselines should be introduced to enable a more rigorous and convincing comparison with the proposed method.
2. According to the results in Table 2, the proposed method shows limited improvement on relatively simple tasks. On the 32B model, no performance gains are observed on either the GSM8K or MATH datasets.

---

> ### Author Rebuttal · Authors · 2025-07-31
>
> Thank you for your constructive comments. We have carefully considered your suggestions and would like to provide the following clarifications.
>
> > **Weakness 1:** To the best of my knowledge, there has been extensive research on effective reasoning. However, this work includes only a limited number of relatively weak baselines. Stronger and more diverse baselines should be introduced to enable a more rigorous and convincing comparison with the proposed method. As mentioned in the Weakness, could the authors consider including additional and stronger baselines?
>
> **Re:** Thank you for your valuable suggestions. We have incorporated additional baselines focused on efficient reasoning. These include training-free methods including Chain-of-Draft [1], Self-Doubt[2] model merge[3], as well as training-based methods CoT-Valve [4] and O1-Pruner [5].
>
> For Chain-of-Draft and Self-Doubt, we followed the original prompts and reasoning strategies, applying them to our LIMO SFT-trained model to evaluate performance. For the Model Merging and CoT-Valve baselines, we selected the highest-performing variants reported in their respective original papers. Specifically, for Model Merging, we employed Ties-Merging to combine our LIMO SFT-trained model with the original instruct model, following the methodology outlined in [3]. For CoT-Valve, we adopted the CoT-Valve+P training approach and utilized their publicly available MixChain-Z dataset for training, as described in [4]. During evaluation, we sampled each question 16 times and computed the average performance across all methods to ensure fair and consistent comparisons. The results, summarized in the table below, demonstrate that QFFT exhibits strong competitiveness across various metrics.
>
> Additionally, QFFT and long-to-short methods are orthogonal. This means that the Long CoT outputs from QFFT models can be further compressed using long-to-short techniques, thereby elevating the upper limit of QFFT's efficiency. We provide experiments demonstrating this in the table below.
>
> | Method | GSM8K Acc ↑ | GSM8K Tokens ↓ | GSM8K AES ↑ | MATH Acc ↑ | MATH Tokens ↓ | MATH AES ↑ | AIME25 Acc ↑ | AIME25 Tokens ↓ | AIME25 AES ↑ | Avg. Acc ↑ | Avg. Tokens ↓ | Avg. AES ↑ |
> | --- | --- | --- | --- | --- | --- | --- | --- | --- | --- | --- | --- | --- |
> | LIMO 7B (base) | 88.2 | 1.8K | - | 80.4 | 5.9K | - | 16.8 | 17.8K | - | 61.8 | 8.5K | - |
> | Training Free baselines |  |  |  |  |  |  |  |  |  |  |  |  |
> | Chain-of-Draft | 84.2 | 1.5K | -2.9 | 77.1 | 5.6K | -3.6 | 15.7 | 17.6K | -6.4 | 59.0 | 8.23K | -4.3 |
> | Self-Doubt | 87.7 | 1.1K | 3.4 | 79.4 | 4.0K | 1.7 | 15.9 | 12.3K | 2.2 | 61.0 | 5.8K | 2.4 |
> | Model-Merge(Ties-Merging) | 85.0 | 1.1K | 0.3 | 77.0 | 4.1K | -1.2 | 12.7 | 15.9K | -23.3 | 58.23 | 7.03K | -8.1 |
> | Taining-based Methods |  |  |  |  |  |  |  |  |  |  |  |  |
> | CoT-Valve(CoT-Valve+P-MixChain-Z) | 82.7 | 0.5K | 1.0 | 66.2 | 3.1K | -12.9 | 14.4 | 12.8K | -11.5 | 54.43 | 5.47K | -7.8 |
> | O1-pruner | 90.8 | 0.8K | 5.8 | 78.2 | 3.2K | 1.8 | 14.2 | 12.2K | -12.7 | 61.07 | 5.4K | -1.7 |
> | QFFT (Ours) | 88.0 | 0.7K | 5.9 | 80.6 | 4.1K | 2.9 | 17.2 | 15.6K | 1.4 | 61.93 | 6.8K | 3.4 |
> | QFFT + SimPO | 89.0 | 0.5K | 7.3 | 80.2 | 3.4K | 4.1 | 17.7 | 13.6K | 2.9 | 62.3 | 5.83K | 4.8 |
>
> [1] Chain of Draft: Thinking Faster by Writing Less
>
> [2] Revisiting Overthinking in Long Chain-of-Thought from the Perspective of Self-Doubt
>
> [3] Unlocking Efficient Long-to-Short LLM Reasoning with Model Merging
>
> [4] CoT-Valve: Length-Compressible Chain-of-Thought Tuning
>
> [5] O1-Pruner: Length-Harmonizing Fine-Tuning for O1-Like Reasoning Pruning
>
> > **Weakness 2:** According to the results in Table 2, the proposed method shows limited improvement on relatively simple tasks. On the 32B model, no performance gains are observed on either the GSM8K or MATH datasets.
>
> **Re:** Thank you for your question. Indeed, as shown in Table 2 in our paper, QFFT provides limited efficiency improvements (AES) on relatively simple evaluation datasets (such as GSM8K), especially when compared to strong long-to-short baselines like O1-Pruner. This is because QFFT is fundamentally an adaptive reasoning method: its efficiency gains come from replacing Long CoT with Short CoT for simple questions, rather than compressing the Long CoT responses. In contrast, long-to-short methods compress the redundant parts of Long CoT responses.
>
> However, long-to-short methods have a notable drawback: they can significantly harm performance on more difficult problems. As discussed in work [6][7], adequate reasoning length is critical for "slow thinking" models to effectively solve complex tasks, and compressing the Long CoT chain often leads to severe performance degradation on such tasks. In this regard, QFFT avoids this pitfall and is able to match SFT performance on challenging problems.
>
> Additionally, **QFFT and long-to-short methods are orthogonal**. This means that the Long CoT outputs from QFFT models can be further compressed using long-to-short techniques, thereby elevating the upper limit of QFFT's efficiency. We provide supporting experiments in the table below:
>
> | Method | GSM8K Acc ↑ | GSM8K Tokens ↓ | GSM8K AES ↑ | MATH Acc ↑ | MATH Tokens ↓ | MATH AES ↑ | AIME25 Acc ↑ | AIME25 Tokens ↓ | AIME25 AES ↑ |
> | --- | --- | --- | --- | --- | --- | --- | --- | --- | --- |
> | LIMO 7B (base) | 88.2 | 1.8K | - | 80.4 | 5.9K | - | 16.8 | 17.8K | - |
> | O1-pruner | 90.8 | 0.8K | 5.8 | 78.2 | 3.2K | 1.8 | 14.2 | 12.2K | -12.7 |
> | QFFT (Ours) | 88.0 | 0.7K | 5.9 | 80.6 | 4.1K | 2.9 | 17.2 | 15.6K | 1.4 |
> | QFFT + SimPO | 89.0 | 0.5K | 7.3 | 80.2 | 3.4K | 4.1 | 17.7 | 13.6K | 2.9 |
>
> [6] DAST: Difficulty-Adaptive Slow-Thinking for Large Reasoning Models
>
> [7] s1: Simple test-time scaling.
>
>
> > **Question 2:** How does the proposed method perform on other models, such as LLaMA? It would be helpful to include additional experiments to demonstrate the generalizability of the method across different model architectures.
>
> **Re:** Thank you for your insightful suggestion. In Appendix C.2 (Table 6), we include experiments on the Phi-4-mini-Instruct model, which validate the effectiveness of QFFT on the Phi-4 architecture.
>
> Additionally, we have added experiments on the LLaMA3 architecture, as shown below:
>
> | Data | MATH500 (Acc/Tokens/TAK) | GSM8K (Acc/Tokens/TAK) | AIME25 (Acc/Tokens/TAK) |
> | --- | --- | --- | --- |
> | Llama3.1-8B-Instruct | 44.2/1.1K/0 | 67.4/0.8K/0 | 0.6/2.4K/0 |
> |+ 1k S1.1 QFFT | 56.1/4.8K/28.6 | 76.5/1.3K/26.1 | 4.4/16.5K/9.9 |
> |+ 1k S1.1 SFT | 56.3/9.7K/4.4 | 75.2/5.1K/2.1 | 3.7/24.8K/0 |
> |+ Bespoke-Stratos-17k QFFT | 68.1/3.8K/36.8 | 86.8/1.2K/27.2 | 12.3/13.7K/22.4 |
> |+ Bespoke-Stratos-17k SFT | 68.8/7.1K/0.4 | 86.3/4.0K/3.8 | 12.5/19.6K/0 |
>
> We observe that for LLaMA3.1-8B, both SFT and QFFT perform poorly on AIME25 with only 1k (S1.1) data. We believe this is because 1k samples are insufficient for the LLaMA model to effectively learn Long CoT capabilities. In contrast, with BS-17k data, there is a substantial improvement over the 1k results. We attribute this to characteristics inherent to LLaMA3 itself. Inspired by works [8] and [9], we note that LLaMA3 lacks sufficient injection of Long CoT patterns such as reflection and backtracking, during its pre-training phase (compared to Qwen), thus requiring more data to learn Long CoT patterns effectively.
>
> [8] OctoThinker: Mid-training Incentivizes Reinforcement Learning Scaling
>
> [9] Cognitive Behaviors that Enable Self-Improving Reasoners, or, Four Habits of Highly Effective STaRs
>
>
> > **Question 3:** Since the proposed fine-tuning method removes human user queries, it may potentially weaken the model’s instruction-following ability. Could the authors include experiments on instruction-following benchmarks to rule out this possibility?
>
> **Re:** Thank you for your valuable question. We have conducted experiments on two general instruction-following datasets (IFEval [10] and AlignBench-v1 [11]), as shown below:
>
> | Model | Configuration | IFEval All | IFEval Short CoT (Score / Percentage) | IFEval Long CoT (Score / Percentage) | AlignBench-v1 All | AlignBench-v1 Short CoT (Score / Percentage) | AlignBench-v1 Long CoT (Score / Percentage) |
> | --- | --- | --- | --- | --- | --- | --- | --- |
> | Qwen2.5-7B-Instruct | - | 72.9 | 80.9 / 0% | - | 7.13 | 7.13 / 100%| 0 |
> | Qwen2.5-7B-Instruct | + S1.1 1K SFT | 62.8 | 68.4 / 70.4% | 49.4 / 29.6% | 5.33 | 5.93 / 60.2% | 4.43 / 40.8% |
> | Qwen2.5-7B-Instruct | + S1.1 1K QFFT | 69.3 | 71.0 / 96.2% | 51.2 / 1.5% | 6.06 | 6.12 / 96.9% | 4.28 / 3.1% |
> | Qwen2.5-7B-Instruct | + BS 17K SFT | 63.2 | 69.7 / 62% | 52.6 / 38.0% | 5.01 | 5.64 / 50.7% | 4.36 / 49.3% |
> | Qwen2.5-7B-Instruct | + BS 17K QFFT | 71.7 | 71.0 / 98% | 50.3 / 2.0% | 6.03 | 6.05 / 98.4% | 4.63 / 1.6% |
> | Qwen2.5-32B-Instruct | - | 80.9 | 80.9 / 0 | - | 7.13 | 7.13 / 100% | 0 |
> | Qwen2.5-32B-Instruct | + S1.1 1K SFT | 67.4 | 74.0 / 71.7% | 50.8 / 28.3% | 5.60 | 6.08 / 69.7% | 4.49 / 30.3% |
> | Qwen2.5-32B-Instruct | + S1.1 1K QFFT | 75.1 | 79.5 / 93.3% | 52.8 / 1.7% | 6.71 | 6.81 / 95.0% | 4.87 / 5% |
>
> Based on these experiments, we draw the following conclusions:
>
> (1) Long CoT SFT can degrade the model's instruction-following capabilities on general tasks. Similar findings have been noted in work [12].
>
> (2) Models fine-tuned with QFFT exhibit stronger instruction-following abilities compared to those with SFT. This is because SFT leads to over-reliance on Long CoT patterns, which significantly impairs instruction-following. In contrast, QFFT preserves Short CoT patterns in most cases, thereby maintaining better instruction-following performance on general tasks.
>
> [10] [Instruction-Following Evaluation for Large Language Models](https://arxiv.org/abs/2311.07911)
>
> [11] AlignBench: Benchmarking Chinese Alignment of Large Language Models
>
> [12] Scaling Reasoning, Losing Control: Evaluating Instruction Following in Large Reasoning Models

---

> > ### Comment · Reviewer_CTuE · 2025-08-03
> > **Official Comment by Reviewer CTuE**
> >
> > Thank you for the detailed responses. They addressed my concerns well, especially the additional experiments related to LLaMA. I have raised my score and confidence accordingly.

---

> > > ### Author Response · Authors · 2025-08-03
> > >
> > > Thank you for your valuable feedback and for helping us improve the quality of our paper!

---

### Official Review · Reviewer_dV3j · 2025-07-03

**Clarity:** 3
**Significance:** 3
**Originality:** 3
**Rating:** 5
**Confidence:** 3

**Summary:**

The paper proposes a novel approach to solve the problem of "overthinking" in reasoning models. They train the model only on the answer reasoning chains removing the original question. This results in using shorter reasoning chains for simpler problems and longer for more complex ones.

**Questions:**

Please see the weaknesses

**Ethical Concerns:**

["NO or VERY MINOR ethics concerns only"]

**Limitations:**

yes

**Quality:**

3

**Strengths And Weaknesses:**

Strengths
- The paper proposes a novel approach that targets the problem of overthinking by eliminating the input question while finetuning.
- Despite significant reduction in number of output tokens, QFFT does not degrade the performance much.
- QFFT is able to switch between long and short CoT depending upon the complexity of problem.
- QFFT achieves superior performance on out-of-domain tasks, demonstrating that question free finetuning results in more transferable.


Weaknesses:
- The paper could benifit from a deeper manual analysis of failure modes of QFFT compared to other methods.
- It is not clear whether QFFT would generalize to toher types of reasoning like commmon-sense, scientific etc.
- Does QFFT lead to any reduction in other model capabilities?

---

> ### Author Rebuttal · Authors · 2025-07-31
>
> Thank you for your constructive comments. We have carefully considered your suggestions and would like to provide the following clarifications.
>
> >  **Weakness 1:** The paper could benifit from a deeper manual analysis of failure modes of QFFT compared to other methods.
>
> **Re**: Thanks for your insightful suggestions. To identify the failure modes of QFFT, we carried out experiments on pathological scenarios where QFFT might underperform.
>
> **(1)  We first performed QFFT training in Noisy Scenarios.**
>
> To assess QFFT's robustness to noisy data—common in real-world Long CoT distillation (e.g., incomplete reasoning, incorrect conclusions, irrelevant responses)—we tested it across four noise levels using the LIMO dataset and Qwen2.5-7B-Instruct, evaluating on MATH with 16 runs per question. We aligned the hyperparameters across all settings and, due to resource constraints, trained each model for 3 epochs.
>
> Level I (normal data) served as the baseline. Levels II–IV introduced errors with gradually increasing damage to the data: Level II involved incorrect conclusions, Level III included truncated reasoning, and Level IV contained mismatched question-answer pairs, respectively. As shown in the table below:
>
> | Data   | Method | MATH Pass@1(%) |
> |--------|--------|----------------|
> | Level1 | SFT    | 76.54          |
> | Level1 | QFFT   | 78.61          |
> | Level2 | SFT    | 68.03          |
> | Level2 | QFFT   | 74.69          |
> | Level3 | SFT    | 37.39          |
> | Level3 | QFFT   | 76.89          |
> | Level4 | SFT    | 0.41           |
> | Level4 | QFFT   | 78.61          |
>
>  SFT performance plummeted from 76.5% (Level I) to 0.4% (Level IV), losing reasoning ability with extreme noise. QFFT, however, remained robust, retaining 78.6% performance at Level IV—matching its Level I results—demonstrating clear advantages in handling noisy training data.
>
> **(2)  We also conducted experiments in Low-Resource Scenarios.**
>
> To examine QFFT's performance in low-resource settings, we sampled 10 data points from the S1.1 dataset, with each generating 10 responses via Deepseek-R1 to form 100 training instances. Both SFT and QFFT were trained on these data for 50 epochs with identical parameters. As shown in the table below:
>
> | Training Steps       | SFT Accuracy (%) | QFFT Accuracy (%) |
> |----------------------|------------------|-------------------|
> | 35                   | 71.95            | 76.45             |
> | 70                   | 75.85            | 78.75             |
> | 105                  | 76.85            | 79.15             |
> | 140                  | 74.20            | 78.20             |
> | 175                  | 74.25            | 79.00             |
> | 210                  | 75.05            | 77.75             |
> | 245                  | 74.35            | 76.50             |
> | 280                  | 74.40            | 77.95             |
> | 300（50 epoch）      | 76.00            | 76.75             |
>
> QFFT consistently outperforms SFT across all checkpoints in this low-resource setting. SFT models over-relied on insufficiently internalized Long CoT patterns, limiting performance. In contrast, QFFT retained original Short CoT patterns and adaptively used Long CoT when needed, integrating reflective behaviors with foundational abilities to achieve better results.
>
> **While the aforementioned experiments did not reveal obvious shortcomings of QFFT, theoretically, our method may have a limitation: it cannot be trained on base models (e.g., Qwen2-7B-Base). This is because QFFT requires the model to inherently possess basic Short CoT capabilities, which is why our existing experiments need to use instruction-tuned models (e.g., Qwen2.5-7B-Instruct). We look forward to subsequent researchers with interest exploring this direction more deeply.**
>
>
> >  **Weakness 2:** It is not clear whether QFFT would generalize to other types of reasoning like commmon-sense, scientific etc.
>
> **Re**:   Thank you for your valuable question. We rebuttal from the following two aspects:
>
> (1) Training with mathematical data (LIMO) and evaluating on OOD tasks: In Section 4.4 (Table 4) of our paper, we evaluated performance on out-of-domain tasks, including the general task MMLU-Pro and the scientific task GPQA. The specific results are as follows:
>
>   | Model  | MMLU-Pro | GPQA  |
>   |---|----------|-------|
>   | Qwen-2.5-Instruct-7B        | 40.0     | 36.4  |
>   | &nbsp;&nbsp;+ LIMO-SFT      | 29.3     | 43.2  |
>   | &nbsp;&nbsp;+ LIMO-QFFT     | 48.4     | 44.2  |
>   | Qwen-2.5-Instruct-32B       | 62.2     | 49.5  |
>   | &nbsp;&nbsp;+ LIMO-SFT      | 52.4     | 65.3  |
>   | &nbsp;&nbsp;+ LIMO-QFFT     | 73.4     | 67.9  |
>
> The conclusion is that models trained with QFFT outperform SFT on out-of-domain tasks, including general and scientific reasoning. This demonstrates the superior out-of-domain generalization ability of the QFFT approach.
>
> (2) We also conducted experiments **training** on the Bespoke-Stratos-17k dataset, which includes a mix of mathematical, scientific, and code-related data. Our Table 1 (in our manuscript) includes the results for models trained on Bespoke-Stratos-17k data, where QFFT matches SFT in performance while significantly reducing token count and improving efficiency. This demonstrates that QFFT can also be effectively trained on other types of reasoning data.
>
> >  **Weakness 3:** Does QFFT lead to any reduction in other model capabilities?
>
> **Re**:   This is a question well worth exploring.
>
>   (1) First, like other SFT-based Long reasoning models, QFFT may slightly harm the model's instruction-following capabilities (compared to the base model). We systematically evaluated QFFT and SFT against the base models on instruction-following benchmarks, as shown in the table below:
>
> | Model | Configuration | IFEval All | IFEval Short CoT (Score / Percentage) | IFEval Long CoT (Score / Percentage) | AlignBench-v1 All | AlignBench-v1 Short CoT (Score / Percentage) | AlignBench-v1 Long CoT (Score / Percentage) |
> |-|---|---|--|-|-|-|-|
> | Qwen2.5-7B-Instruct | -| 72.9| 80.9 / 0| - | 7.13 | 7.13 / 100%  | 0|
> | Qwen2.5-7B-Instruct| + S1.1 1K SFT | 62.8| 68.4 / 70.4% | 49.4 / 29.6% | 5.33 | 5.93 / 60.2%  | 4.43 / 40.8% |
> | Qwen2.5-7B-Instruct | + S1.1 1K QFFT  | 69.3  | 71.0 / 96.2%  | 51.2 / 1.5% | 6.06  | 6.12 / 96.9%   | 4.28 / 3.1% |
> | Qwen2.5-7B-Instruct | + BS 17K SFT | 63.2 | 69.7 / 62%| 52.6 / 38.0% | 5.01 | 5.64 / 50.7%  | 4.36 / 49.3%  |
> | Qwen2.5-7B-Instruct | + BS 17K QFFT| 71.7| 71.0 / 98% | 50.3 / 2.0% | 6.03 | 6.05 / 98.4% | 4.63 / 1.6% |
> | Qwen2.5-32B-Instruct | -  | 80.9 | 80.9 / 0 | -  | 7.13 | 7.13 / 100% | 0  |
> | Qwen2.5-32B-Instruct | + S1.1 1K SFT| 67.4 | 74.0 / 71.7%  | 50.8 / 28.3% | 5.60  | 6.08 / 69.7% | 4.49 / 30.3% |
> | Qwen2.5-32B-Instruct| + S1.1 1K QFFT  | 75.1 | 79.5 / 93.3%  | 52.8 / 1.7% | 6.71 | 6.81 / 95.0% | 4.87 / 5% |
>
> But our experimental results indicate that QFFT models exhibit stronger instruction-following capabilities than SFT. On one hand, this highlights the inherent limitations of Long CoT in instruction-following, a point also mentioned in [1]. On the other hand, compared to SFT, QFFT preserves Short CoT patterns, resulting in better instruction-following performance overall.
>
> **We have not observed degradation in other model capabilities from QFFT training. We will continue to monitor and investigate this aspect in future work.**
>
> [1] Scaling Reasoning, Losing Control: Evaluating  Instruction Following in Large Reasoning Models

---

### Official Review · Reviewer_rhbK · 2025-07-03

**Clarity:** 3
**Significance:** 3
**Originality:** 3
**Rating:** 4
**Confidence:** 4

**Summary:**

The paper addresses the efficiency problem in current long chain-of-thought (CoT) reasoning models, which, while strong on complex tasks, often suffer from overthinking—producing unnecessarily verbose reasoning even for simple problems. The authors propose Question-Free Fine-Tuning (QFFT), a distillation approach where the model is fine-tuned only on reasoning chains (CoT responses), with the input questions removed during training. This approach aims to preserve concise short-CoT reasoning for easy problems while enabling adaptive invocation of long-CoT (reflective) reasoning only when necessary. Experimental results on a range of mathematical benchmarks (e.g., GSM8K, MATH, AIME) and out-of-domain tasks show that QFFT reduces average response length by over 50% while maintaining comparable performance to standard SFT, and demonstrates better generalization and adaptivity. A new metric, Thinking Adaptive Cohen’s Kappa (TAK), is introduced to quantify the adaptivity in reasoning pattern selection.

**Questions:**

1. Have the authors considered or evaluated hybrid fine-tuning schemes, such as partial masking of questions, curriculum masking, or re-introducing questions at inference time? Would this further improve adaptivity or mitigate risks of instruction-following degradation?
2. Is there any observed degradation in instruction-following or user prompt adherence, especially for non-mathematical tasks, due to removal of questions during training? If so, can the authors comment or provide measurements?

**Ethical Concerns:**

["NO or VERY MINOR ethics concerns only"]

**Final Justification:**

The results in the original paper and rebuttal are very impressive, although I am still a bit suspicious on how this can work this well, I want to maintain my positive score to reflect solely based on the objective results.

**Limitations:**

yes

**Quality:**

3

**Strengths And Weaknesses:**

Strengths:
1. The paper identifies the important yet overlooked problem of overthinking in long-CoT distilled models.
2. QFFT, which removes the input question in fine-tuning, is a straightforward method.
3. Across multiple mathematical reasoning datasets, QFFT matches or slightly exceeds SFT on accuracy, while dramatically reducing token usage.
4. The introduction of the TAK metric and detailed analyses (e.g., how QFFT models switch to long-CoT patterns only when needed) provides clear evidence for the method’s claimed adaptivity.

Weaknesses:
1. The main contribution is empirical and conceptual, not algorithmic.
2. While the paper analyzes adaptivity, it could benefit from a more in-depth ablation, e.g., testing hybrid or partial question masking, or examining pathological cases where QFFT may fail.
3. The discussion of potential negative impacts (e.g., removing questions during training could risk instruction-following degradation in some applications) is not fully explored in the main paper, though some limitations are covered in the appendix.

---

> ### Author Rebuttal · Authors · 2025-07-31
>
> Thank you for your constructive comments. We have carefully considered your suggestions and would like to provide the following clarifications.
>
> > **Weakness 1**: The main contribution is empirical and conceptual, not algorithmic.
>
> **Re**:   Thank you for your feedback. We acknowledge that the QFFT method may appear relatively simple. However, our approach is highly meaningful and provides valuable insights.
>
> (1) First, QFFT is a simple yet highly effective method. It only requires removing the input question from the training data, with the rest of the process identical to standard SFT. Additionally, QFFT matches SFT's performance while significantly improving token efficiency (reducing average response length by over 50%) and achieving better generalization to out-of-domain tasks.
>
> (2) QFFT offers a novel methodological insight: it enables the model to learn new patterns without overwriting existing ones, while ensuring the model adaptively employs the new patterns under its default behavior. This provides a foundational framework for flexibly incorporating additional patterns.
>
> (3) We provide a probabilistic explanation of why QFFT achieves adaptive reasoning:
>
> The core goal of QFFT is to simultaneously achieve **preserving Short CoT capability** and **triggering Long CoT for challenging problems**, and this mechanism can be intuitively explained through conditional probability:
>
> - The original model already possesses the capability of generating concise reasoning (Short CoT) for a question $Q$, corresponding to the conditional probability of Short CoT inference $P(\text{Short CoT} \mid Q)$. Unlike SFT, which forces the model to learn the direct mapping from questions to Long CoT as $P(\text{Long CoT} \mid Q)$, QFFT avoids overwriting the original $P(\text{Short CoT} \mid Q)$ by removing the question $Q$ from training data. This ensures the model defaults to Short CoT patterns during inference.
>
> - Meanwhile, Long CoT response data inherently contains patterns like reflection (denoted as $B_r$), which emerge when encountering uncertainties or errors. Specifically, during training, QFFT enables the model to learn Long CoT reasoning capabilities and the conditional probability of reflective behaviors. Formally, let $U_L$ denote uncertainties or errors arising in Long CoT reasoning; the model learns: $P_\theta(B_r \mid U_L)$, where $B_r$ represents the occurrence of reflective behaviors, and $P_\theta$ denotes the model-parameterized conditional probability distribution.
>
> - During inference, the model defaults to Short CoT. When Short CoT fails to solve challenging problems (accompanied by uncertainties or errors), denoted as $U_S$, the model learned $P_\theta (B_r \mid U_L)$ in the QFFT training. This enables the activation of $B_r$ when $U_S$ occurs, thereby triggering Long CoT for correction.
>
>
> > **Weakness 2**:  While the paper analyzes adaptivity, it could benefit from a more in-depth ablation, e.g., testing hybrid or partial question masking, or examining pathological cases where QFFT may fail.
>
> **Re**: We provide detailed clarification to your questions in our response to **Weakness 3** and **Question 1**.
>
>
> > **Weakness 3**: The discussion of potential negative impacts (e.g., removing questions during training could risk instruction-following degradation in some applications) is not fully explored in the main paper, though some limitations are covered in the appendix.
>
> **Re**:  Thanks for your insightful suggestions.   We conducted experiments on pathological scenarios where QFFT may fail, focusing on two key scenarios: Noisy Scenarios and Low-Resource Scenarios. **While our experiments did not reveal performance degradation of QFFT relative to SFT in these scenarios, we acknowledge that investigating more diverse scenarios would be valuable and plan to explore them as future work.**
>
> **(1) We first performed QFFT training in Noisy Scenarios.**
>
> To assess QFFT's robustness to noisy data—common in real-world Long CoT distillation (e.g., incomplete reasoning, incorrect conclusions, irrelevant responses)—we tested it across four noise levels using the LIMO dataset and Qwen2.5-7B-Instruct, evaluating on MATH with 16 runs per question. We aligned the hyperparameters across all settings and, due to resource constraints, trained each model for 3 epochs.
>
> Level I (normal data) served as the baseline. Levels II–IV introduced errors with gradually increasing damage to the data: Level II involved incorrect conclusions, Level III included truncated reasoning, and Level IV contained mismatched question-answer pairs, respectively. As shown in the table below:
>
> | Data   | Method | MATH Pass@1(%) |
> |---|--|-|
> | Level1 | SFT | 76.54|
> | Level1 | QFFT | 78.61|
> | Level2 | SFT | 68.03|
> | Level2 | QFFT | 74.69  |
> | Level3 | SFT  | 37.39|
> | Level3 | QFFT | 76.89 |
> | Level4 | SFT  | 0.41  |
> | Level4 | QFFT | 78.61  |
>
> SFT performance plummeted from 76.5% (Level I) to 0.4% (Level IV), losing reasoning ability with extreme noise. QFFT, however, remained robust, retaining 78.6% performance at Level IV—matching its Level I results—demonstrating clear advantages in handling noisy training data.
>
> **(2) We also conducted experiments in Low-Resource Scenarios.**
>
> To examine QFFT's performance in low-resource settings, we sampled 10 data points from the S1.1 dataset, with each generating 10 responses via Deepseek-R1 to form 100 training instances. Both SFT and QFFT were trained on these data for 50 epochs with identical parameters. As shown in the table below:
>
> | Training Steps  | SFT Accuracy (%) | QFFT Accuracy (%) |
> |--|--|-|
> | 35| 71.95| 76.45|
> | 70 | 75.85 | 78.75 |
> | 105 | 76.85| 79.15 |
> | 140| 74.20| 78.20|
> | 175| 74.25 | 79.00|
> | 210| 75.05 | 77.75 |
> | 245 | 74.35| 76.50 |
> | 280| 74.40 | 77.95 |
> | 300（50 epoch）| 76.00 | 76.75|
>
> QFFT consistently outperforms SFT across all checkpoints in this low-resource setting.
>
> > **Question 2**: Is there any observed degradation in instruction-following or user prompt adherence, especially for non-mathematical tasks, due to removal of questions during training? If so, can the authors comment or provide measurements?
>
> **Re**:  Thank you for your valuable question.  We have conducted experiments on two general instruction-following datasets (IFEval [1] and AlignBench-v1 [2]), as shown below:
>
> | Model | Configuration | IFEval All | IFEval Short CoT (Score / Percentage) | IFEval Long CoT (Score / Percentage) | AlignBench-v1 All | AlignBench-v1 Short CoT (Score / Percentage) | AlignBench-v1 Long CoT (Score / Percentage) |
> |-|---|---|--|-|-|-|-|
> | Qwen2.5-7B-Instruct | -| 72.9| 80.9 / 0| - | 7.13 | 7.13 / 100%  | 0|
> | Qwen2.5-7B-Instruct| + S1.1 1K SFT | 62.8| 68.4 / 70.4% | 49.4 / 29.6% | 5.33 | 5.93 / 60.2%  | 4.43 / 40.8% |
> | Qwen2.5-7B-Instruct | + S1.1 1K QFFT  | 69.3  | 71.0 / 96.2%  | 51.2 / 1.5% | 6.06  | 6.12 / 96.9%   | 4.28 / 3.1% |
> | Qwen2.5-7B-Instruct | + BS 17K SFT | 63.2 | 69.7 / 62%| 52.6 / 38.0% | 5.01 | 5.64 / 50.7%  | 4.36 / 49.3%  |
> | Qwen2.5-7B-Instruct | + BS 17K QFFT| 71.7| 71.0 / 98% | 50.3 / 2.0% | 6.03 | 6.05 / 98.4% | 4.63 / 1.6% |
> | Qwen2.5-32B-Instruct | -  | 80.9 | 80.9 / 0 | -  | 7.13 | 7.13 / 100% | 0  |
> | Qwen2.5-32B-Instruct | + S1.1 1K SFT| 67.4 | 74.0 / 71.7%  | 50.8 / 28.3% | 5.60  | 6.08 / 69.7% | 4.49 / 30.3% |
> | Qwen2.5-32B-Instruct| + S1.1 1K QFFT  | 75.1 | 79.5 / 93.3%  | 52.8 / 1.7% | 6.71 | 6.81 / 95.0% | 4.87 / 5% |
>
> Based on these experiments, we draw the following conclusions:
>
> - Long CoT SFT can degrade the model's instruction-following capabilities on general tasks. Similar findings have been noted in work [3].
>
> - Models fine-tuned with QFFT exhibit stronger instruction-following abilities compared to SFT. This is because SFT leads to relying more on Long CoT patterns, which significantly impairs instruction-following. In contrast, QFFT preserves Short CoT patterns in most cases, thereby maintaining better instruction-following performance on general tasks.
>
> [1] Instruction-Following Evaluation for Large Language Models
>
> [2] AlignBench: Benchmarking Chinese Alignment of Large Language Models
>
> [3] Scaling Reasoning, Losing Control: Evaluating  Instruction Following in Large Reasoning Models
>
> > **Question 1**: Have the authors considered or evaluated hybrid fine-tuning schemes, such as partial masking of questions, curriculum masking, or re-introducing questions at inference time? Would this further improve adaptivity or mitigate risks of instruction-following degradation?
>
> **Re**:   Thank you for your insightful suggestions. This is a point well worth exploring.
>
> We conducted hybrid question masking experiments: **progressively incorporating samples with questions into the QFFT training process, which, from the perspective of SFT, is equivalent to progressively masking questions.**  We have supplemented our work with the following experiment: For the S1.1 1k dataset, we progressively masked the questions in α% of the samples and then performed SFT. Subsequently, we evaluated the trained model on the MATH500 dataset. We tracked changes in adaptivity (TAK) and the proportion of Short CoT usage as the ratio of samples with questions increased, as shown in the table below:
>
> | Ratio(α) of Data with Question (%) | Ratio of Short CoT (%) | TAK  |
> |-|-|-|
> | 0%| 56.2| 38.4|
> | 0.1% | 40.6| 30.5|
> | 1% | 13.1| 8.8|
> | 10% | 10.2 | 8.4|
> | 100%| 8.0 | 6.2 |
>
> This experiment leads to the following conclusions: As the proportion of masked questions increases(fewer samples with questions), the model's adaptivity (TAK) steadily improves.  This partially validates the hypothesis in Section 2.3 of our paper: Even an extremely small proportion of samples with question (eg. 1%) mixed into the data can override the model's original mapping to Short CoT responses, erasing the model’s prior ability to respond concisely. This demonstrates the rationality of the motivation behind the QFFT method.

---

> > ### Comment · Reviewer_rhbK · 2025-08-05
> >
> > Thanks authors for the rebuttal response. These results are extremely good. To be honest I can't wrap my head around how QFFT is not influenced by the added noise at all, especially that it maintains the exact same performance (78.61) while SFT baseline drops to 0.41. I can't seem to understand how removing questions would magically make the model not need to train with longer steps, or improve instruction following capabilities. Would you please provide more intuition on why removing questions would fundamentally provide so much benefits? Thanks.

---

> ### Author Response · Authors · 2025-08-05
>
> Thanks for your feedback and questions. We provide the following clarifications.
>
> >**Q1: Explanation of QFFT's Robustness in Noisy Scenarios**
>
> **Re**: The reason QFFT is less affected by noise lies in the distinct impact mechanisms of the designed noise levels on SFT and QFFT:
>
> (1) These noise levels disrupt the correct "question-to-response (Q→R) mapping" that SFT relies on during training. SFT will learn the association between specific questions and their corresponding reasoning processes, but noise (such as incorrect conclusions, incomplete reasoning, or irrelevant question-answer pairs) directly corrupts this mapping. As a result, the model learns erroneous associations, which explains the drastic performance drop of SFT.
>
>
>
> (2) However, the noise does not completely destroy the reasoning structures contained in the responses. Even in noisy data, response sequences still retain partial or complete reasoning steps (e.g., mathematical deductions, logical verifications). This allows SFT to maintain some performance under mild noise and, more importantly, provides a stable learning foundation for QFFT. QFFT focuses on learning reasoning patterns from responses rather than relying on Q→R mappings, thus remaining largely unaffected by noise.
>
> Particularly in the Level IV scenario (QFFT remains 78.61 while SFT baseline drops to 0.41), where the Q→R mapping is completely erroneous, SFT is forced to learn these incorrect associations, leading to the acquisition of flawed reasoning processes and eventual performance collapse. In contrast, QFFT, by completely removing questions during training, consistently learns from the reasoning structures within responses—its training process is identical to that in Level I (normal data) . This design prevents it from being impacted by the misalignment between questions and reasoning, enabling stable performance retention.
>
> >**Q2: Why QFFT Achieves Better Performance with Fewer Training Steps in Low-Resource Scenarios Compared to SFT**
>
> **Re**: When training with a limited number of steps (e.g., 35 steps), the model's Long CoT capabilities are not yet well learned. However, QFFT models continue to utilize short CoT patterns for many questions, and their inherent capability in solving tasks like MATH500 remains unaffected.  Consequently, QFFT demonstrates better performance compared to SFT at the same step count, as SFT primarily relies on long CoT patterns.
>
> >**Q3: Why QFFT Improves Instruction-Following Capabilities Compared to SFT**
>
> **Re**: Models fine-tuned with QFFT exhibit stronger instruction-following abilities compared to those with SFT. This is because SFT encourages over-reliance on long CoT patterns, which significantly impairs instruction-following. (Long CoT SFT can degrade the model's instruction-following capabilities on general tasks, as similar findings have been noted in prior work [1].) In contrast, QFFT preserves short CoT patterns in most cases. Since the base model's short CoT capabilities (Qwen2.5-Instruct) inherently support strong instruction-following, QFFT overall maintains superior instruction-following performance compared to SFT.
>
> [1] Scaling Reasoning, Losing Control: Evaluating Instruction Following in Large Reasoning Models

---

> ### Author Response · Authors · 2025-08-08
>
> Dear Reviewer rhbK,
>
> Thank you very much for your valuable feedback. We hope the clarifications provided have addressed your questions adequately.
>
> If there are any remaining points that need further clarification, please feel free to let us know—we're happy to address them further.
>
> Thank you again for your time and insights.
>
> Best regards
>
> The Authors

---

### Official Review · Reviewer_wE8y · 2025-07-03

**Clarity:** 3
**Significance:** 3
**Originality:** 3
**Rating:** 4
**Confidence:** 2

**Summary:**

This work introduces a method to train the reasoning models (by removing the Question while training) such that the method now generates less steps (less number of tokens)  while not degrading the accuracy much as compared to baseline. They do it in Supervised Fine Tuning Regime.

**Questions:**

(1) Qwen is trained on short COT and now you are trying to make it adapt to long COT to make it think more and improve performance on difficult questions. To do this you distill long answers from Deepseek R1 and use it in SFT training and QFFT training (removing Q from the training). Intuitively by SFT training one would assume that the model becomes better at long COT. Because of being trained on long answers SFT will automatically learn to generate larger tokens.  If we remove the question and do QFFT training, its seems to me that the length of each datapoint has become smaller and hence the model will learn to generate lesser token as compared to SFT. It will be interesting to me if we can compare:
1. Qwen base model's token count for gsm8k with that of QFFT model.
2. Qwen base model's token count for AIME25 with that of QFFT model.


(II) Gains over strong long-to-short baselines (e.g., O1-Pruner, SimPO) are not too much. Is it possible to provide a more detailed justification?

**Ethical Concerns:**

["NO or VERY MINOR ethics concerns only"]

**Final Justification:**

My questions were resolved fully by the rebuttal especially the intuitive explanation and additional generalizability results. Hence I maintained my score of leaning towards accept.

**Quality:**

3

**Strengths And Weaknesses:**

Strengths:
1. The method is original.
2. The method significantly reduces the token count while being trained on small number of samples only. The method don't really benefit much by training on larger number of samples (BS-17k vs S1.1).
3. Extensive experiments are provided on in-domain and OOD as well SOTA Qwen model.

Weaknesses:
1. While the method is original, I am struggling to get an intuitive understanding of why the method works ? And why removing Q from the training does not affect performance ?
2. Table 1 should include a row of actual base model's performance to understand how the training data affects SFT and QFFT. When directly checking the results on Qwen model, I see 3%-4% drop. Doesn't that signify that the training data is not useful for the model training and decrease performance.
3. As the data difficultly increases, accuracy drop of QFFT as compared to SFT also becomes larger.
4. This work has tested its method on one series of model which is Qwen. To be sure of the method's generalizability it would be more convincing if the method was tested on other series of model too which follows short COT.

Typos:
1. Line 195: setup of, ... (missing citation)
2. Line 286:  Is it achieves ?

---

> ### Author Rebuttal · Authors · 2025-07-30
>
> Thank you for your constructive comments. We have carefully considered your suggestions and would like to provide the following clarifications.
>
> > **Strength 1**: The method significantly... The method don't really benefit much by training on larger number of samples (BS-17k vs S1.1).
>
> **Re**:    Thank you for your recognition of our work and your valuable questions. Regarding the question: **our method does not really benefit much by training on larger number of samples (BS-17k vs. S1.1)**, we provide the following explanations:
>
>   (1) In Table 1 of our paper, we compared the performance of QFFT and SFT on both the BS-17k and S1.1 datasets. QFFT achieves results comparable to SFT on both datasets. As for the lack of greater improvement on S1.1 compared to BS-17k, **we believe this may be attributed to the inherent quality of the training data, rather than a limitation of our QFFT method.**
>
>   (2) As noted in Paper [1], just 1K high-quality samples from S1.1 can achieve effects comparable to the full 59K dataset. Therefore, although S1.1 has fewer samples, its superior quality is sufficient to endow the model with strong Long CoT capabilities, yielding performance close to that of the BS-17k dataset.
>
>   [1] s1: Simple test-time scaling.
>
> > **Weakness 1**: While the method is original, I am struggling to get an intuitive understanding of why the method works? And why removing Q from the training does not affect performance?
>
> **Re**:    The core goal of QFFT is to simultaneously achieve **preserving Short CoT capability** and **triggering Long CoT for challenging problems**, and this mechanism can be intuitively explained through conditional probability:
>
> (1) The original model already possesses the capability of generating concise reasoning (Short CoT) for a question $Q$, corresponding to the conditional probability of Short CoT inference $P(\text{Short CoT} \mid Q)$. Unlike SFT, which forces the model to learn the direct mapping from questions to Long CoT as $P(\text{Long CoT} \mid Q)$, QFFT avoids overwriting the original $P(\text{Short CoT} \mid Q)$ by removing the question $Q$ from training data. This ensures the model defaults to Short CoT patterns during inference.
>
> (2) Meanwhile, Long CoT response data inherently contains patterns like reflection (denoted as $B_r$), which emerge when encountering uncertainties or errors. Specifically, during training, QFFT enables the model to learn Long CoT reasoning capabilities and the conditional probability of reflective behaviors. Formally, let $U_L$ denote uncertainties or errors arising in Long CoT reasoning; the model learns: $P_\theta(B_r \mid U_L)$, where $B_r$ represents the occurrence of reflective behaviors, and $P_\theta$ denotes the model-parameterized conditional probability distribution.
>
> During inference, the model defaults to Short CoT. When Short CoT fails to solve challenging problems (accompanied by uncertainties or errors), denoted as $U_S$, the model learned $P_\theta (B_r \mid U_L)$ in the QFFT training. This enables the activation of $B_r$ when $U_S$ occurs, thereby triggering Long CoT for correction.
>
>   > **Weakness 2**: Table 1 should include a row of actual base model's performance. When directly checking the results on Qwen model, I see 3%-4% drop. Doesn't that signify that the training data is not useful for the model training and decrease performance.
>
> **Re**: Thank you for your valuable question. We have added the performance of the base models for comparison, as shown in the table below:
>
>   | Model | GSM8K | MATH500 | AIME25 |
>   |-|-|-|-|
>   | Qwen2.5-7B-Instruct   | 90.7  | 74.8 | 9.8  |
>   | S1.1-SFT-7B | 90.6 | 80.8    | 18.2 |
>   | S1.1-QFFT-7B | 91.0  | 80.2 | 17.2 |
>   | Qwen2.5-32B-Instruct  | 93.6  | 82.8  | 17.5   |
>   | S1.1-SFT-32B | 92.8  | 93.1 | 48.6 |
>   | S1.1-QFFT-32B| 93.6  | 92.2 | 46.8   |
>
>   (1) Our reported results are slightly lower (by about 2%) compared to those in the Qwen official blog [2]. This is because we report pass@16 values (averaging over 16 random samplings), whereas the official Qwen results use greedy sampling once. As a result, our metrics are more robust.
>
>   (2) On the MATH500 and AIME25 datasets, both SFT and QFFT show clear improvements over the base models. For simpler datasets like GSM8K, the advantages of Long CoT over Short CoT are not as pronounced as they are on more challenging datasets (e.g., AIME25).
>
>   [2] Qwen2.5-LLM: Extending the boundary of LLMs
>
>   > **Weakness 3**: As the data difficultly increases, accuracy drop of QFFT as compared to SFT also becomes larger.
>
> **Re**: Thank you for your insightful question. However, we believe that the observed larger accuracy drop as data difficulty increases is due to random fluctuations.
>
> (1) The AIME25 dataset consists of only 30 questions, which introduces significant randomness and  high variability (although run at 16 times).
>
> (2) In Appendix C.1 Table 5 (Supplementary Material), we report the performance of QFFT on AIME24 and AMC, as shown below:
>
> | Data   | Method | AMC | AIME24 |
> |-|-|-|-|
> | S1.1   | SFT | 56.1 | 19.0 |
> | S1.1   | QFFT | 55.3    | 20.6  |
> | S1.1   | Δ | -0.8 | +1.6  |
> | LIMO   | SFT | 55.8| 19.1  |
> | LIMO   | QFFT | 57.2    | 19.6 |
> | LIMO   | Δ | +1.4 | +0.5 |
> | BS-17K | SFT | 61.6 | 19.4|
> | BS-17K | QFFT | 61.6  | 20.6  |
> | BS-17K | Δ | 0.0  | +1.2 |
>
> On the challenging datasets AIME24 and AMC, QFFT performs better than SFT on average. Therefore, we believe that QFFT can match the performance of SFT on both difficult and simple problems.
>
>   > **Weakness 4**: This work has tested its method on Qwen. To be sure of the method's generalizability it would be more convincing if the method was tested on other series of model.
>
> **Re**:  Thank you for your insightful suggestion. In Appendix C.2 (Table 6), we include experiments on the Phi-4-mini-Instruct model, which validate the effectiveness of QFFT on the Phi-4 architecture. Additionally, we have added experiments on the LLaMA3 architecture, as shown below:
>
> | Data | MATH500 (Acc/Tokens/TAK) | GSM8K  (Acc/Tokens/TAK)| AIME25  (Acc/Tokens/TAK)|
> |-|-|-|-|
> | Llama3.1-8B-Instruct| 44.2/1.1K/0 | 67.4/0.8K/0 | 0.6/2.4K/0  |
> | + 1k S1.1 QFFT |  56.1/4.8K/28.6 | 76.5/1.3K/26.1 | 4.4/16.5K/9.9 |
> | + 1k S1.1 SFT|  56.3/9.7K/4.4| 75.2/5.1K/2.1| 3.7/24.8K/0|
> | + Bespoke-Stratos-17k QFFT | 68.1/3.8K/36.8  | 86.8/1.2K/27.2 | 12.3/13.7K/22.4  |
> |+ Bespoke-Stratos-17k SFT  |  68.8/7.1K/0.4 | 86.3/4.0K/3.8 | 12.5/19.6K/0  |
>
> We observe that for LLaMA3.1-8B, both SFT and QFFT perform poorly on AIME25 with only 1k (S1.1) data. We believe this is because 1k samples are insufficient for the LLaMA model to effectively learn Long CoT capabilities. In contrast, with BS-17k data, there is a substantial improvement over the 1k results. We attribute this to characteristics inherent to LLaMA3 itself. Inspired by works [3] and [4], we note that LLaMA3 lacks sufficient injection of Long CoT patterns such as reflection and backtracking, during its pre-training phase (compared to Qwen), thus requiring more data to learn Long CoT patterns effectively.
>
> [3] OctoThinker: Mid-training Incentivizes  Reinforcement Learning Scaling
>
>  [4] Cognitive Behaviors that Enable Self-Improving Reasoners, or, Four Habits of Highly Effective STaRs
>
>   > **Question 1**:  Qwen is trained on short COT and now you are trying to make it adapt to... 1. Qwen base model's token count for gsm8k with that of QFFT model. 2. Qwen base model's token count for AIME25 with that of QFFT model.
>
> **Re**:   Thank you for your thoughtful insights. The table below compares the accuracy and token counts for the Qwen base model, QFFT on GSM8K and AIME25:
>
>   | Model | GSM8K (Acc/Tokens) | AIME25 (Acc/Tokens)  |
>   |-|-|-|
>   | Qwen2.5-7B-Instruct   | 90.7 / 436  | 9.8 / 1865|
>   | + S1.1 QFFT | 91.0 / 443 | 17.2 / 12789 |
>   | + S1.1 SFT   | 90.6 / 1766  | 18.2 / 17740 |
>
>   But we think your understanding of QFFT needs to be clarified.
>
> (1) Long CoT responses are typically much longer than the question lengths (e.g., distilled DeepSeek-R1 responses for MATH and AIME can reach 5k or more than 10k tokens). Compared to the response, the question length is almost negligible.
>
> (2) The reason QFFT works is shown in our response to your **Weakness 1**.
>
>   > **Question 2**: Gains over strong long-to-short baselines (e.g., O1-Pruner, SimPO) are not too much. Is it possible to provide a more detailed justification?
>
> **Re**: Thank you for your valuable question.
>
> (1) While QFFT achieves efficiency comparable to strong long-to-short baselines, it has unique advantages. Its training is exceptionally simple: based on SFT, it only removes input queries, needing no extra data or complex reinforcement learning (unlike O1-Pruner), thus reducing training overhead. Moreover, QFFT outperforms in areas beyond long-to-short methods' scope, As shown in Section 4.4 (Table 3) of our paper, it exhibits better hallucination mitigation on LLM-AggreFact and superior out-of-domain generalization (e.g., on GPQA and MMLU-Pro), granting it distinct value.
>
> (2) QFFT and long-to-short methods are orthogonal. This means that the Long CoT outputs from QFFT models can be further compressed using long-to-short techniques, thereby elevating the upper limit of QFFT's efficiency. We provided experiments demonstrating this in the table below.
>
> | Method | GSM8K Acc ↑ | GSM8K Tokens ↓ | GSM8K AES ↑ | MATH Acc ↑ | MATH Tokens ↓ | MATH AES ↑ | AIME25 Acc ↑ | AIME25 Tokens ↓ | AIME25 AES ↑ |
> |-|-|-|-|-|-|-|-|-|-|
> | LIMO 7B (base)   | 88.2 | 1.8K   | -    | 80.4 | 5.9K  | -    | 16.8 | 17.8K | -   |
> | O1-pruner  | 90.8  | 0.8K | 5.8   | 78.2  | 3.2K | 1.8| 14.2 | 12.2K   | -12.7 |
> | QFFT (Ours)  | 88.0 | 0.7K  | 5.9  | 80.6 | 4.1K | 2.9 | 17.2 | 15.6K | 1.4 |
> | QFFT + SimPO | 89.0  | 0.5K   | **7.3** | 80.2| 3.4K  | **4.1** | 17.7| 13.6K | **2.9** |

---

> > ### Comment · Reviewer_wE8y · 2025-08-05
> >
> > Thanks for the nice rebuttal. I have decided to maintain my score.

---

> > > ### Author Response · Authors · 2025-08-06
> > >
> > > Thank you for your valuable feedback and for helping us improve the quality of our paper!

---

> ### Author Response · Authors · 2025-08-04
>
> Dear reviewer wE8y，
>
> Thank you again for your thoughtful review and feedback, which we have carefully addressed above.
>
> We believe these clarifications fully address your main concerns and would appreciate it if you could indicate whether they sufficiently resolve your points and might positively impact your score. If not, we welcome further discussion to better understand any remaining issues.
>
> Thank you again for your time and consideration.
>
> Best regards,
>
> The Authors

---

### Decision · Program_Chairs · 2025-09-17

**Decision:**

Accept (spotlight)

**Comment:**

This paper introduces Question-Free Fine-Tuning (QFFT), a simple yet novel approach to mitigating the inefficiency of long chain-of-thought (CoT) reasoning in large language models. Instead of training on question–answer pairs, QFFT fine-tunes solely on reasoning chains, encouraging models to adaptively balance concise reasoning on simple problems and reflective reasoning on harder ones. Experiments on multiple benchmarks suggest that QFFT reduces token usage by over 50% while maintaining comparable or slightly improved accuracy, with some evidence of stronger generalization to out-of-domain tasks.

All reviewers agree that the method is conceptually simple, original, and effective, addressing the important and underexplored issue of overthinking in long-CoT reasoning models. Comprehensive experiments demonstrate that QFFT significantly improves token efficiency while preserving or modestly improving task accuracy. The introduction of the Thinking Adaptive Cohen’s Kappa (TAK) metric provides a useful lens for measuring adaptive reasoning. The paper is clearly written, well structured, and supported by both in-domain and out-of-domain evaluations. Reviewers highlight that the rebuttal satisfactorily addressed concerns regarding generalizability to additional model families and potential degradation of instruction-following.

On the other hand, several reviewers point out that the set of baselines is limited. Comparisons against stronger and more diverse long-to-short reasoning approaches (e.g., SimPO, pruning-based methods) would strengthen the empirical case. Some concerns remain about generalizability, as the evaluation focuses primarily on Qwen, with limited results on other architectures, though the rebuttal provided additional supporting evidence. A deeper analysis of failure modes, hybrid schemes (partial question masking), or potential impacts on instruction following could make the work more robust.

Overall, this paper makes a meaningful and original contribution to adaptive reasoning in large language models. While not without limitations, the simplicity and effectiveness of QFFT, together with the strong empirical evidence, make it a valuable addition to the literature. I recommend acceptance.